# Bi-perspective Splitting Defense: Achieving Clean-Seed-Free Backdoor Security

**Yangyang Shen**[1]  **Xiao Tan**[1]  **Dian Shen**[1]  **Meng Wang**[2]  **Beilun Wang**[1]

## Abstract

Backdoor attacks have seriously threatened deep neural networks (DNNs) by embedding concealed vulnerabilities through data poisoning. To counteract these attacks, training benign models from poisoned data garnered considerable interest from researchers. High-performing defenses often rely on additional clean subsets/seeds, which is untenable due to increasing privacy concerns and data scarcity. In the absence of additional clean subsets/seeds, defenders resort to complex feature extraction and analysis, resulting in excessive overhead and compromised performance. To address these challenges, we identify the key lies in sufficient utilization of both the easier-to-obtain target labels and clean hard samples. In this work, we propose a Bi-perspective Splitting Defense (BSD). BSD distinguishes clean samples using both semantic and loss statistics characteristics through open set recognition-based splitting (OSS) and altruistic model-based data splitting (ALS) respectively. Through extensive experiments on benchmark datasets and against representative attacks, we empirically demonstrate that BSD surpasses existing defenses by over 20% in average Defense Effectiveness Rating (DER), achieving clean data-free backdoor security.

## 1. Introduction

Recent studies exposed the vulnerabilities of deep neural networks (DNNs) to various attacks (Carlini & Wagner, 2017; Moosavi-Dezfooli et al., 2016; Kurakin et al., 2018; Zeng et al., 2019; Ilyas et al., 2018), among which backdoor attacks (Li et al., 2022; Wenger et al., 2021; Zhang et al., 2021; Wang et al., 2020) have emerged as a significant threat due to their ease of execution and profound impact. Owing to their non-model-manipulation property and congruence with actual model training scenarios, data poisoning-based backdoor attacks (Goldblum et al., 2022; Shafahi et al., 2018) stand out as prevalent and impactful threats, highlighting the importance of backdoor defense research. Taking facial recognition as an example (Figure 1), poisoned data may induce DNNs to erroneously learn a strong correlation between the adversary-defined trigger pattern (e.g., sunglasses) and the target label (e.g., a high-authority individual). While behaving normally without the trigger, the backdoored model predicts any individuals wearing sunglasses as the predetermined high-authority person.

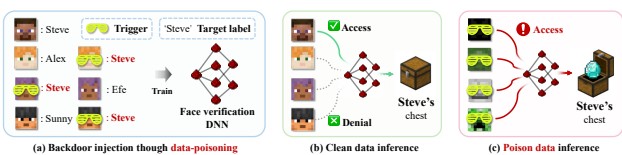

*Figure 1.* Illustration of data-poisoning-based backdoor attacks.

Recently, a branch of in-training defenses has focused on training benign models directly from poisoned data, which is particularly significant when developing our own models using untrustworthy datasets. They primarily adhere to a data-splitting paradigm that differentiates between benign and poisoned samples, and disrupts the association between trigger patterns and target labels to mitigate backdoor behaviors (Li et al., 2021a; Huang et al., 2022; Gao et al., 2023). However, these defenses either highly rely on impractical clean subsets/seeds(i.e., additional clean data outside the training set; hereafter referred to as "clean subsets" or "clean seeds") or have unsatisfactory performance due to limited defensive perspective.

Clean subsets have proven effective in various backdoor defenses (Zhu et al., 2024; Liu et al., 2018; Wu & Wang, 2021; Zeng et al., 2021; Li et al., 2023a; Gao et al., 2023), as they provide insights into benign samples. However, recollecting a clean subset can be prohibitively expensive, especially when the training set contains numerous classes (e.g., acquiring new benign facial records for millions of individuals in a facial recognition database). Additionally, manually inspecting a large training set to identify a clean subset is both labor-intensive and raises significant privacy concerns. Seemingly effective, some methods expand the clean subset under the premise that an additional clean set

---

[1]School of Computer Science and Engineering, Southeast University, Nanjing, China [2]College of Design and Innovation, Tongji University, Shanghai, China. Correspondence to: Beilun Wang <beilun@seu.edu.cn>.

*Proceedings of the 42nd International Conference on Machine Learning*, Vancouver, Canada. PMLR 267, 2025. Copyright 2025 by the author(s).

is available (Pan et al., 2023), which still contradicts real-world scenarios.

Existing potential clean data acquisition methods, however, yield only a very small clean subset and still rely on downstream defenses (Zeng et al., 2023a). This exacerbates the already large hyperparameter search space. Moreover, in the event of potential failures, the presence of evaded malicious samples can erode the effectiveness of the clean subset and ruin the defense pipeline.

In practical scenarios where no clean seed samples are available, existing in-training defenses that do not effectively utilize the semantic and loss statistics perspectives are struggling to distinguish poisoned samples. They involve complicated feature extraction and analysis, suffering from significant training costs (Huang et al., 2022) and compromised performance (Li et al., 2021a; Chen et al., 2022a; Tran et al., 2018; Weber et al., 2023; Liu et al., 2023).

These constraints bring us back to the core issue: ***How to eliminate dependence on impractical clean subsets while maintaining both the efficiency and effectiveness of defenses?***

In this work, we focus on improving the state-of-the-art in-training defense under the practical yet challenging non-clean-seed-involved scenario. We identify that existing methods have failed to fully utilize the readily available target labels and clean hard samples, leading to incomplete exploitation of the information inherent in the poisoned dataset. By integrating an open set recognition game into a refined loss-guided split, we propose a Bi-perspective Splitting Defense (BSD). Specifically, BSD introduces open set recognition-based splitting (OSS) and altruistic model-based data splitting (ALS). OSS reframes the identification of poisoned samples within the target class as an open set recognition problem and identify poison samples semantically. ALS utilizes an altruistic model to reveal reliable clean hard samples with high loss values. These two mechanisms complement each other by leveraging distinct judgment perspectives, the intersection of their results provides a robust clean pool.

Through extensive experiments on benchmark datasets and against representative attacks, we empirically demonstrate that BSD surpasses existing defenses by over 20% in average Defense Effectiveness Rating (DER), achieving clean-data-free backdoor security.

## 2. Related works

Currently, backdoor defenses fall into two main categories:

**Post-training backdoor defenses** focus on repairing a backdoored model with a set of locally prepared clean training sets. Trigger inversion (Sur et al., 2023) is a popular method

to reconstruct the trigger pattern and then unlearn it to renovate the model. In addition to trigger-synthesis defenses, pruning, distillation, finetuning, and model connectivity analysis (Liu et al., 2018; Wu & Wang, 2021; Li et al., 2023a) are widely applied in the realm of backdoor defense as well. Despite the promising results, most post-training methods assume using an extra clean set for defense, which introduces potential limitations.

**In-training backdoor defenses** aim at training a benign model from the polluted dataset, which holds considerable practical significance (Chen et al., 2022a; Tran et al., 2018; Weber et al., 2023; Liu et al., 2023). Following an intuitive idea of splitting the dataset into clean and poison pools and treating them separately, several representative training-time defenses, namely Anti-backdoor learning (ABL) (Li et al., 2021a), Decoupled-based defense (DBD) (Huang et al., 2022), and Adaptive splitting-based defense (ASD) (Gao et al., 2023), have garnered attention. Anti-backdoor learning (ABL) (Li et al., 2021a) isolates a small ratio of poisoned samples through local gradient ascent and unlearns these samples to neutralize the effect of remaining poisoned samples in the clean pool. Decoupling-based defense (DBD) (Huang et al., 2022) utilizes self-supervised learning to acquire a benign feature extractor and uses a clean subset to initialize the classifier head. Then, it separates the suspicious according to the loss magnitude and breaks the link between the trigger and the target label through semi-supervised learning. Adaptive splitting-based defense (ASD) (Gao et al., 2023) further improves the initialization based on clean seed samples and introduces meta-split to identify clean hard samples, achieving higher clean accuracy (CA). Besides these defenses, adopting differential-privacy SGD (Du et al., 2019) and strong data augmentation (Borgnia et al., 2021) can also defend against backdoor attacks to some degree. Our BSD belongs to the data-splitting in-training defenses and makes further adaptions.

## 3. Preliminaries

### 3.1. Threat model

Following (Gao et al., 2023), We adopt the poisoning-based threat model used in previous works (Gu et al., 2017; Chen et al., 2017; Turner et al., 2018), where the training dataset contains a set of pre-crafted poisoned samples provided by attackers. As a typical setting of training-time defenses in previous works (Gao et al., 2023; Borgnia et al., 2021; Du et al., 2019; Huang et al., 2022; Li et al., 2021a), we assume that defenders have control over the training process.

### 3.2. Problem formulation

The malicious training set from the adversaries can be denoted as $\mathcal{D} = \mathcal{D}_c \cup \mathcal{D}_p$, where $\mathcal{D}_c$ and $\mathcal{D}_p$ are two disjoint

part of the raw benign dataset $\mathcal{D}_{\text{raw}} = \{(x_i, y_i)\}_{i=1}^N$. Each $x_i \in \mathcal{X} \subset \mathbb{R}^{C \times W \times H}$. The ground-truth labels $y_i \in \mathcal{Y} = \{0, 1, \ldots, C-1\}$, with $C$ being the number of categories. Given the poisoning rate $\rho$, $\mathcal{D}_c$ has $(1-\rho)N$ samples. The poisoned set $\mathcal{D}_p = \{(G(x), T(y)) \mid (x, y) \in \mathcal{D}_{\text{raw}} \backslash \mathcal{D}_c\}$, where $G : \mathcal{X} \to \mathcal{X}, T : \mathcal{Y} \to \mathcal{Y}$ are the attack-specific poisoned image generator and label modifier. As an example, $G(x) = m \odot x + (1-m) \odot t, T(x) = y_t$, where the mask $m \in \{0, 1\}^{C \times W \times H}, t \in \mathcal{X}$ is the trigger pattern, and $y_t$ is the target label. We call the $\{(x,y)|y \neq y_t, (x,y) \in \mathcal{D}\}$ as non-target samples $\mathcal{D}_{\text{nt}}$, and $\{(x,y)|y = y_t, (x,y) \in \mathcal{D}\}$ as target samples $\mathcal{D}_{\text{t}}$, $\{(x,y)|y = y_t, (x,y) \in \mathcal{D}_c\}$ as clean target samples $\mathcal{D}_{\text{ct}}$. Note that under our clean-seed-free scenario, defenders have no knowledge of which portions of the training set $\mathcal{D}$ are benign and which are not, nor do they have access to any additional clean samples as external clean subsets or seeds.

Following the natural idea to exclude the poison samples from the training set, defenders can divide $\mathcal{D}$ into a clean pool $\mathcal{D}_{\tilde{c}}$ and a poison pool $\mathcal{D}_{\tilde{p}}$. To prevent the model from being backdoored while preserving the performance on benign samples, the core is breaking the link between triggers and target labels, and making the best of the poison pool. We follow DBD and ASD to use semi-supervised learning (Berthelot et al., 2019b) that only leverages visual features of samples in the poison pool:

$$\mathcal{L}_{\text{semi}} = \sum_{(x,y) \in \mathcal{D}_{\tilde{c}}} \mathcal{L}_s(x, y; \theta) + \lambda \sum_{x \in \mathcal{D}_{\tilde{c}} \backslash \mathcal{D}_{\tilde{p}}} \mathcal{L}_u(x; \theta), \quad (1)$$

where $\theta$ denote the weights of the main model $f(x; \theta)$ ($f_\theta$ for simplicity), $\mathcal{L}_s$ is a common supervised loss function such as cross-entropy loss, the unsupervised $\mathcal{L}_u$ is applied on the suspicious polluted set $\mathcal{D}_{\tilde{c}} \backslash \mathcal{D}_{\tilde{p}}$, with a trade-off coefficient $\lambda$. Appendix B.6 provides a detailed definition of semi-supervised learning.

The main task under this framework lies in finding an appropriate indicator that helps maximize the difference between benign and poisoned samples, thus returning a clean pool with high precision and a poison pool with high recall, i.e.:

$$\min_{\mathcal{D}_{\tilde{c}}} |\mathcal{D}_p \cap \mathcal{D}_{\tilde{c}}| \text{ s.t. } \mathcal{D}_{\tilde{c}} \subset \mathcal{D}, \ \max_{\mathcal{D}_{\tilde{p}}} |\mathcal{D}_p \cap \mathcal{D}_{\tilde{p}}| \text{ s.t. } \mathcal{D}_{\tilde{p}} \subset \mathcal{D}. \quad (2)$$

### 3.3. Open set recognition

There are four basic recognition categories of classes in Open set recognition (Geng et al., 2020): 1) known known classes (KKCs), i.e., the classes with distinctly labeled positive training samples (also serving as negative samples for other KKCs), and even have the corresponding side-information like semantic/attribute information, etc; 2) known unknown classes (KUCs), i.e., labeled negative samples, not necessarily grouped into meaningful classes, such

as the background classes, the universum classes, etc; 3) unknown known classes (UKCs), i.e., classes with no available samples in training, but available side information (e.g., semantic/attribute information) of them during training; 4) unknown unknown classes (UUCs), i.e., classes without any information regarding them during training: not only unseen but also having not side information (e.g., semantic/attribute information, etc.) during training.

## 4. Proposed method

Our BSD has three main components as illustrated in Figure 2. As we assume no extra clean subset/seeds access, pool initialization is vital to the defense. To ensure a secure initialization, open set recognition-based splitting (OSS) and altruistic model-based splitting (ALS) focus on the perspectives of image **semantic information** and **loss statistics** respectively. Based on the altruistic model introduced in ALS, we further improve the pool update with class completion and selective dropping strategy.

**(1) OSS** is motivated by the similarity between the open set recognition task and poison sample detection in backdoor defense. As the main model is warmed up using $\mathcal{D}_{\text{nt}}$, poison samples are unknown-known-classes (UKCs) whose semantic information is included in $\mathcal{D}_{\text{nt}}$, thus having smaller minimum distances to feature clusters of known-known-classes (KKCs). Clean target samples fall into a new cluster and have larger minimum distances. Detailed description in Section 4.1.1.

**(2) ALS** highlights the clean hard samples with high loss values in the altruistic model, which could filter out the overfitted poison samples. A detailed description of ALS is provided in Section 4.1.2.

**(3) Subsequent training** of BSD follow a loss-guided split, which uses the loss difference of a sample between the main and altruistic model to distinguish samples. BSD compensates the less selected categories and drops the evaded poison samples using class completion and selective dropping strategies respectively. A detailed description of subsequent training is provided in Section 4.2.

The detailed algorithm is presented in Appendix A.

### 4.1. The initialization of clean and poison pools

The initialization of the clean and poison pool is then obtained by intersecting the consensual clean samples in ALS and OSS:

$$\mathcal{D}_{\tilde{c}} = \mathcal{D}_{c_{als}} \cap \mathcal{D}_{c_{oss}}, \ \mathcal{D}_{\tilde{p}} = \mathcal{D}_{p_{als}} \cup \mathcal{D}_{p_{oss}}, \quad (3)$$

where the $\mathcal{D}_{c_{als}}$ and $\mathcal{D}_{p_{als}}$ is the split result of ALS, $\mathcal{D}_{c_{oss}}$ and $\mathcal{D}_{p_{oss}}$ is the split result of OSS. The following two subsections will explain the two initialization mechanisms.

*Figure 2.* An overview of our BSD. BSD consists of two main initialize mechanisms, i.e., open set recognition-based splitting (OSS) and altruistic model-based splitting (ALS). BSD use the intersection of $\mathcal{D}_{c_{oss}}$ and $\mathcal{D}_{c_{als}}$ from OSS and ALS as the clean pool initialization. In the next two stages of the subsequent training, BSD dynamically updates the clean and poison pools based on a loss-guided split strategy based on the loss discrepancy of the main model $f_\theta$ and the altruistic model $g_\varphi$. The pseudo-code of BSD is provided in Appendix A.

### 4.1.1. OPEN SET RECOGNITION BASED SPLITING

Open set recognition (OSR) is a task that aims to accurately identify known classes while also recognizing or rejecting unknown classes when the input may contain both. In the context of OSR, identifying unknown-known classes (UKCs) and unknown-unknown classes (UUCs) are two major tasks. Here UKCs refer to classes that have no available samples during training, but their side information (such as semantic/attribute information, etc.) can be obtained during training. UUCs refer to classes that do not have any relevant information during the training process: not only have they not been seen, but there is also no side information during the training process.

We notice that distinguishing the clean target samples and poison samples is related to the UKCs and UUCs identification in OSR. The poison samples are sort of UKCs because the triggers do not corrupt their semantic informa-

tion. Hence, we set out to cast the clean target samples to UUCs, which can reframe the splitting within the target class into an OSR problem.

To make the poison samples and clean target samples belong to the UKCs and UUCs respectively, the known-known classes (KKCs, i.e. the training set) should contain the semantic classes of UKCs ($\mathcal{D}_p$), while information of the UUCs ($\mathcal{D}_{ct}$) is not included. Therefore, we construct the KKCs with the non-target classes ($\mathcal{D}_{nt}$) which satisfies both requirements above. Thus, we can train the main model $f_\theta$ on $\mathcal{D}_{nt}$ for its warm-up, i.e., $\theta = \arg\min \mathcal{L}_{\text{semi}}(\mathcal{D}_{nt}; f_\theta)$.

Now the local detection of poisoned samples in $\mathcal{D}_t$ has been reframed as an open set recognition problem. The clean pool identified by OSS can be acquired by adding the approximated UUCs ($\mathcal{D}_{\tilde{ct}}$) to the KKCs ($\mathcal{D}_{nt} = \{(x,y)|y \neq$

$y_t, (x, y) \in \mathcal{D}\}$):

$$\mathcal{D}_{c_{\mathrm{oss}}} = \mathcal{D}_{\tilde{\mathrm{ct}}} \cup \mathcal{D}_{\mathrm{nt}}, \ \mathcal{D}_{p_{\mathrm{oss}}} = \mathcal{D} \backslash \mathcal{D}_{c_{\mathrm{oss}}}. \quad (4)$$

To approximate the $\mathcal{D}_{\tilde{\mathrm{ct}}}$ in the reframed problem, it's ideal to have the known-unknown classes (KUCs), which again indicates the need for clean seed samples. Fortunately, there have been a lot of previous studies on solving this problem without KUCs. We approximate $\mathcal{D}_{\tilde{\mathrm{ct}}}$ by:

$$\mathcal{D}_{\tilde{\mathrm{ct}}} = \left\{ (x, y) \mid \mathcal{S}(x) \geq \mathrm{Percentile}\left(\mathcal{D}_t^{\mathcal{S}}, 1 - \beta\right) \right\}, \quad (5)$$

where $\mathrm{Percentile}(Set, ratio)$ returns the $ratio$-percentile in $Set$, $\beta$ is a fixed ratio of samples in $\mathcal{D}_t$ to be added to $\mathcal{D}_{\mathrm{nt}}$, $\mathcal{D}_t^{\mathcal{S}}$ denotes the set of score values obtained by applying $\mathcal{S}$ to all samples in $\mathcal{D}_t$, i.e., $\mathcal{D}_t^{\mathcal{S}} = \{\mathcal{S}(x) \mid (x, y) \in \mathcal{D}_t\}$. Motivated by OpenMAX (Bendale & Boult, 2016), we take the feature distance to KKCs as a metric to measure the likelihood of a sample within $\mathcal{D}_t$ to be a true clean sample:

$$\mathcal{S}(x) = \min_{i = \{0,1,\dots,C-1\} \backslash \tilde{y_t}} \left\{ \|f_e(x) - \mu_i\|_2 \right\}, \quad (6)$$

where $f_e$ is the feature extractor of $f$, $\mu_i = \frac{1}{N_i} \sum f_e(x_i)$ is the cluster center of each KKC.

Approximating $y_t$. It should be noted that it requires $y_t$ to construct $\mathcal{D}_t$ and $\mathcal{D}_{\mathrm{nt}}$. Although the target label $y_t$ used in the above process is unknown to the defender, it's easy to approximate. There exist various alternative methods to detect the $y_t$ (Gao et al., 2024; Zhu et al., 2024), Here we are motivated by (Zhu et al., 2024) to use the most frequent second likely prediction, i.e., $y_t = \mathrm{argsort}(\text{-logit})[1]$, where logit means the logit output of a DNN. However, this prediction could be unstable, we further statistics the predicted $y_t$ in each warm-up epoch and use the majority as the final prediction of $y_t$ (this process will be pre-completed in ALS, details in Appendix A).

### 4.1.2. ALTRUISTIC MODEL BASED SPLITTING

In our BSD, we introduce an altruistic model $g(x; \varphi)$ ($g_\varphi$ for simplicity), which is an independent model having the same structure as the main model. It serves as a pathfinder of the main model by exposing itself to the entire malicious training set, i.e., $\varphi = \mathrm{argmin} \mathcal{L}_{\mathrm{ce}}(\mathcal{D}, g_\varphi)$, where $\mathcal{L}_{\mathrm{ce}}$ stands for the cross-entropy loss.

We calculate the rest unsolved part in (3), i.e., $\mathcal{D}_{c_{als}}$ and $\mathcal{D}_{p_{als}}$ following the equation below:

$$\mathcal{D}_{c_{als}} = \left\{ (x, y) \mid \mathcal{L}(x, y, \varphi) \geq \mathrm{Percentile}\left(\mathcal{D}^{\mathcal{L}}, 1 - \alpha\right) \right\},$$
$$\mathcal{D}_{p_{als}} = \mathcal{D} \backslash \mathcal{D}_{c_{als}}, \quad (7)$$

where $\mathcal{L}$ is the symmetric cross-entropy loss (Wang et al., 2019), $\mathcal{D}^{\mathcal{L}} = \{\mathcal{L}(x, y, \varphi) \mid (x, y) \in \mathcal{D}\}$ is loss values using $g_\varphi$ of the training set, $\alpha$ is the ratio of samples split to the clean pool.

Note that although here the altruistic model is just used for the pool initialization, it also plays a significant role in the subsequent training.

## 4.2. Subsequent training

BSD adaptively updates the pools according to the loss discrepancy of $f_\theta$ and $g_\varphi$ in the subsequent training, ensuring balanced and robust learning

Class completion strategy. Despite securing good pool initialization without involving the clean seed samples, the clean pools may have an unbalanced distribution of classes, hampering the model's performance on clean accuracy. This primarily stems from the imbalanced learning status of categories and the cyclic positive feedback effect of loss-guided methods. We further revise the splitting strategy of clean samples, adding samples in the class with the fewest samples:

$$\mathcal{D}_{\tilde{c_1}} = \{(x, y) \mid \mathcal{I}(x, y) \geq \mathrm{Percentile}(\mathcal{D}^{\mathcal{I}}, 1 - \alpha)$$
$$\vee \{y = i, \mathcal{I}(x, y) \geq \mathrm{Percentile}(\mathcal{D}_i^{\mathcal{I}}, 1 - n_i'/N_i)\}\},$$
$$\mathcal{D}_{\tilde{p_1}} = \mathcal{D} \backslash \mathcal{D}_{\tilde{c_1}}, \quad (8)$$

where $\mathcal{I}(x, y)$ is an loss based indicator, $\mathcal{D}^{\mathcal{I}} = \{\mathcal{I}(x, y) \mid (x, y) \in \mathcal{D}\}$ is the mapped $\mathcal{D}$ using $\mathcal{I}$. $\mathcal{D}_i^{\mathcal{I}} = \{\mathcal{I}(x, y) \mid y = i(x, y) \in \mathcal{D}\}$, $N_i = |\mathcal{D}_i^{\mathcal{I}}|$, $n_i' = \min\{\alpha n_i, N_{secondFew}\}$, $N_{secondFew}$ is number of samples in the second-fewest predicted class.

We do subtraction between the loss of samples on the main and altruistic models, as the poison samples should also have high loss values on the unaffected main model and low loss values on the backdoored altruistic model. Thus $\mathcal{I}$ is defined as:

$$\mathcal{I}(x, y) = \mathcal{L}_{sce}(x, y, \varphi) - \mathcal{L}_{sce}(x, y, \theta), \quad (9)$$

where $\mathcal{L}_{sce}$ denotes the symmetric cross-entropy loss (Wang et al., 2019).

Selective dropping strategy. Approaching the end of the training, we drop the samples that are predicted to be $\tilde{y_t}$ by both models:

$$\mathcal{D}_{\tilde{c_2}} = \mathcal{D}_{\tilde{c_1}} \backslash \{(x, y) \mid (f(x) = \tilde{y_t}) \wedge (g(x) = \tilde{y_t})\},$$
$$\mathcal{D}_{\tilde{p_2}} = \mathcal{D} \backslash \mathcal{D}_{\tilde{c_2}}, \quad (10)$$

There exist two probable situations for a sample that will be dropped: 1) the sample is poisoned; 2) the sample is a clean sample with the original label being $\tilde{y_t}$. For situation 1, it is the correct decision to drop poisoned samples; for situation 2, the agreement between the two models indicates the sample is already well-fitted by both models and is less important. As a result, the dropping of these samples generally helps improve model performance.

# 5. Experiments

## 5.1. Experimental settings

**Datasets and DNN models.** We adopt three benchmark datasets for the evaluation of the backdoor defenses, namely, CIFAR-10 (Krizhevsky et al., 2009), GTSRB (Stallkamp et al., 2012), and Imagenet (Deng et al., 2009). The results are conducted with ResNet-18 (He et al., 2016) and MobileNet-v2 (Sandler et al., 2018) as the backbone models for their representativeness and widespread use.

**Attack Baselines.** We implement seven representative attacks, i.e., BadNets (Gu et al., 2017), Blended (Chen et al., 2017), WaNet (Nguyen & Tran, 2021), Label-Consistent(LC) (Turner et al., 2019), ReFool (Liu et al., 2020), SIG (Barni et al., 2019), and Narcissus (Zeng et al., 2023b). All these attacks are implemented based on open-source codebases of ASD (Gao et al., 2023), DBD (Huang et al., 2022), Narcissus (Zeng et al., 2023b), backdoor-Bench (Wu et al., 2022), and BackdoorBox (Li et al., 2023b). The first five attacks follow the same setting in settings in (Gao et al., 2023) unless otherwise specified, SIG and Narcissus follow the setting with (Li et al., 2021a) and (Zeng et al., 2023b) respectively, while the poisoning rate $\rho$ and target label $y_t$ are the same as LC. A detailed description of the attack implementations is provided in Appendix B.3.

**Defense Baselines.** We compare our proposed BSD with five existing backdoor defenses, namely Fine-pruning (FP) (Liu et al., 2018), Neural Attention Distillation (NAD) (Li et al., 2021b), Anti-Backdoor-Learning (ABL) (Li et al., 2021a), Decoupling-based Backdoor Defense (DBD) (Huang et al., 2022), and Adaptive Splitting-based backdoor Defense (ASD) (Gao et al., 2023). The detailed settings for all defense baselines are as suggested in ASD. For our BSD, we adopt the MixMatch (Berthelot et al., 2019b) semi-supervised training framework for the main model, following Decoupling-based Defense (DBD) and Adaptive Splitting-based Defense (ASD). The altruistic model undergoes a warm-up phase with 25 epochs, utilizing the Adam optimizer, cross entropy loss, with a learning rate of 0.001. The default warm-up epochs for the main model in OSS are set to 20 ($T_1 = 20$), with a default fixed $\beta$ of 0.2. Class completion training spans 60 epochs ($T_2 = 90$), and selective dropping training spans 30 epochs ($T_3 = 120$). The clean pool ratio $\alpha$ follows a sinusoidal growth curve during class rebalance training, starts at 0.2, and reaches an upper limit of 0.6 at the end of the class completion stage, after which it remains fixed. Additional details are available in Appendix B.4.

**Evaluation metrics.** We assess the effectiveness of backdoor defenses using two widely used metrics: Clean Accuracy (CA) and the attack success rate (ASR). To be specific, the CA is the accuracy of clean data, the ASR is defined as the proportion of poisoned samples that are misclassified as the target class by the model. In the context of backdoor defense, superior performance is characterized by higher CA and lower ASR. To comprehensively evaluate the performance of defense methods, we include another metric named Defense Effectiveness Rating (DER) (Zhu et al., 2023a), higher DER indicate better defense performance. The detailed definition of DER is provided in Appendix B.5.

## 5.2. Main results

We present a summary of CAs, ASRs, and DERs achieved by five backdoor defenses against three most representative backdoor attacks on three benchmark datasets in Table 1[1]. As illustrated in Table 1, our BSD has the best average DERs on each dataset, being capable of maintaining high CAs without compromising the robustness indicated by ASRs. In comparison with post-training defenses, i.e., FP and NAD, which require thousands of clean seed samples, BSD consistently outperforms them with lower ASRs when OSS is used as the alternative initialization. Additionally, the CAs of BSD surpass those of FP and NAD. Concerning recently proposed training-time defenses, the BSD has best result in general. ABL, which assumes no presence of clean subsets, has relatively close performance under CIFAR-10 & BadNets, GTSRB & BadNets, and GTSRB & Blend. Nevertheless, the CA under CIFAR-10 & WaNet indicates a class underfitting collapse (CAs on certain classes are close to 0%) and its performance is inferior to that of BSD in general. For another representative training-time defense DBD, although it has a slight edge in ASRs on CIFAR-10, its average ASRs and CAs fall behind our BSD. ASD, which assumes an extra small clean seed set is characterized by consistent high CAs and stable ASRs. However, BSD still surpasses it in general. In summary, our BSD performance remains competitive and, in some cases, surpasses that of state-of-the-art methods.

In addition to the representative attacks, we investigated four more attacks that may be threatening to existing defenses. They consist of one invisible attack, ReFool (Liu et al., 2020), and three clean-label attacks, LC (Turner et al., 2019), SIG (Barni et al., 2019), and Narcissus (Zeng et al., 2023b). ReFool uses a physical yet stealthy reflection trigger, which makes the backdoor hard to detect. LC, SIG, and Narcissus belong to the clean-label attack, which is a type of tricky backdoor attack that does not change the label of samples, making most of the defenses ineffective (where DBD has the most significant performance degradation). For our BSD, clean-label attacks are less threatening.

---

[1]Since we strictly follow the same settings, we reference the baseline results for CIFAR-10 and GTSRB from ASD (Gao et al., 2023). However, the exact 30 randomly selected classes from the Imagenet subset used are unknown to us, so we ran all the baselines on Imagenet using our own randomly chosen 30 classes.

*Table 1.* The clean accuracy (CA%), attack success rate (ASR%), and defense effective rating (DER%) of 5 baseline backdoor defense methods and our BSD against 3 representative backdoor attacks on 3 benchmark datasets. The baselines consist of two post-training defenses (FP, NAD) and three state-of-the-art training-time defenses (ABL, DBD, ASD). 'Non' stands for no defense. The best and second best results are in **bold** and underlined.

| DATASET | ATTACK | METRIC | NON | FP | NAD | ABL | DBD | ASD | BSD(OURS) |
|---|---|---|---|---|---|---|---|---|---|
| CIFAR-10 | BADNET | CA | 94.9 | 93.9 | 88.2 | 93.8 | 92.3 | 93.4 | **95.1** |
| | | ASR | 100.0 | 1.8 | 4.6 | 1.1 | **0.8** | 1.2 | 0.9 |
| | | DER | - | 98.6 | 94.4 | **98.9** | 98.3 | 98.7 | 99.6 |
| | BLENDED | CA | 94.1 | 92.9 | 85.8 | 91.9 | 91.7 | 93.7 | **94.9** |
| | | ASR | 98.3 | 77.1 | 3.4 | 1.6 | **0.7** | 1.6 | 0.8 |
| | | DER | - | 60.0 | 93.3 | 97.3 | 97.6 | 98.2 | **98.8** |
| | WANET | CA | 93.6 | 90.4 | 71.3 | 84.1 | 91.4 | 93.1 | **94.5** |
| | | ASR | 99.9 | 98.6 | 6.7 | 2.2 | **0.0** | 1.7 | 0.8 |
| | | DER | - | 49.1 | 85.5 | 94.1 | 98.9 | 98.9 | **99.6** |
| | AVERAGE DER | | - | 69.2 | 91.0 | 96.8 | 98.3 | 98.6 | **99.3** |
| GTSRB | BADNET | CA | 97.6 | 84.2 | 97.1 | 97.1 | 91.4 | 96.7 | **97.6** |
| | | ASR | 100.0 | **0.0** | 0.2 | **0.0** | **0.0** | **0.0** | **0.0** |
| | | DER | - | 93.3 | 99.7 | 99.8 | 96.9 | 99.6 | **100.0** |
| | BLENDED | CA | 97.2 | 91.4 | 93.3 | **97.1** | 91.5 | **97.1** | 96.9 |
| | | ASR | 99.4 | 68.1 | 62.4 | 0.5 | 99.9 | 0.3 | **0.0** |
| | | DER | - | 62.8 | 66.6 | 99.4 | 46.9 | 99.5 | **99.6** |
| | WANET | CA | 97.2 | 92.5 | 96.5 | 97.0 | 89.6 | **97.2** | 97.2 |
| | | ASR | 100.0 | 21.4 | 47.1 | 0.4 | **0.0** | 0.3 | 0.2 |
| | | DER | - | 87.0 | 76.1 | 99.7 | 96.2 | **99.9** | 99.9 |
| | AVERAGE DER | | - | 81.0 | 80.8 | 99.6 | 80.0 | 99.6 | **99.8** |
| IMAGENET | BADNET | CA | 75.7 | 71.4 | 51.7 | 68.1 | 76.1 | **81.1** | 78.3 |
| | | ASR | 99.5 | 2.6 | 2.5 | 7.6 | 1.2 | 100.0 | **1.1** |
| | | DER | - | 96.3 | 86.5 | 92.2 | **99.2** | 50.0 | 99.2 |
| | BLENDED | CA | 74.5 | 73.1 | 42.8 | 61.9 | 77.9 | 79.7 | **80.1** |
| | | ASR | 97.7 | 81.9 | **0.2** | 100.0 | 35.0 | 51.0 | **0.2** |
| | | DER | - | 57.2 | 82.9 | 42.6 | 81.4 | 73.4 | **98.8** |
| | WANET | CA | 77.1 | 76.9 | 74.0 | 74.9 | 77.2 | 78.4 | **78.7** |
| | | ASR | 81.0 | 0.4 | 1.3 | 1.1 | 5.2 | 14.0 | **0.0** |
| | | DER | - | 90.2 | 88.3 | 88.9 | 87.9 | 83.5 | **90.5** |
| | AVERAGE DER | | - | 81.2 | 85.9 | 74.5 | 89.5 | 69.0 | **96.2** |

*Table 2.* The clean accuracy (CA%), attack success rate (ASR%), and defense effective rating (DER%) of 5 baseline backdoor defense methods and our BSD against 4 threatening backdoor attacks on CIFAR-10. The best and second best results are in **bold** and underlined.

| ATTACK | METRIC | NON | FP | NAD | ABL | DBD | ASD | BSD(OURS) |
|---|---|---|---|---|---|---|---|---|
| LC | CA | 94.4 | 87.1 | 85.9 | 80.2 | 83.2 | **93.9** | 92.4 |
| | ASR | 99.9 | 24.4 | 50.5 | 1.6 | 98.1 | 73.2 | **1.2** |
| | DER | - | 84.1 | 70.5 | 92.1 | 45.3 | 63.1 | **98.4** |
| SIG | CA | 95.0 | 87.1 | 85.8 | 67.6 | 80.1 | 93.5 | **93.8** |
| | ASR | 95.2 | 60.8 | 83.0 | 5.1 | 99.9 | 96.5 | **0.0** |
| | DER | - | 63.3 | 51.5 | 81.4 | 42.6 | 49.3 | **97.0** |
| REFOOL | CA | 95.2 | 86.5 | 85.6 | 76.3 | **90.8** | 86.8 | 94.8 |
| | ASR | 99.0 | 23.0 | 42.5 | 82.0 | 2.3 | **0.4** | 0.5 |
| | DER | - | 83.6 | 73.4 | 49.0 | 96.1 | 95.1 | **99.0** |
| NARCISSUS | CA | 95.2 | 87.2 | 86.5 | 79.3 | 87.3 | 93.9 | **94.3** |
| | ASR | 99.5 | 63.4 | 81.0 | 7.1 | 99.5 | **0.0** | **0.0** |
| | DER | - | 64.0 | 54.8 | 88.2 | 46.0 | 99.1 | **99.3** |
| AVERAGE DER | | - | 73.8 | 62.6 | 77.7 | 57.5 | 76.6 | **98.4** |

While the OSS mechanism can be evaded as the semantic information is $\mathcal{D}_t$ is consistent. Fortunately, ALS still functions effectively with its loss-perspective splitting in this scenario, compensating for the limitations of OSS. As shown in Table 2, BSD is not evaded by any of the attacks and achieves the best average DER. Additional details of attack implementation are available in Appendix B.3.

### 5.3. Robustness to different model structures

**BSD makes no assumptions about model structures**, ensuring both compatibility and versatility. To validate this, we evaluated the defense performance of BSD using another widely adopted network, MobileNet (Sandler et al., 2018). As shown in Table 3, BSD consistently outperforms the baseline method with MobileNet-v2 as the backbone.

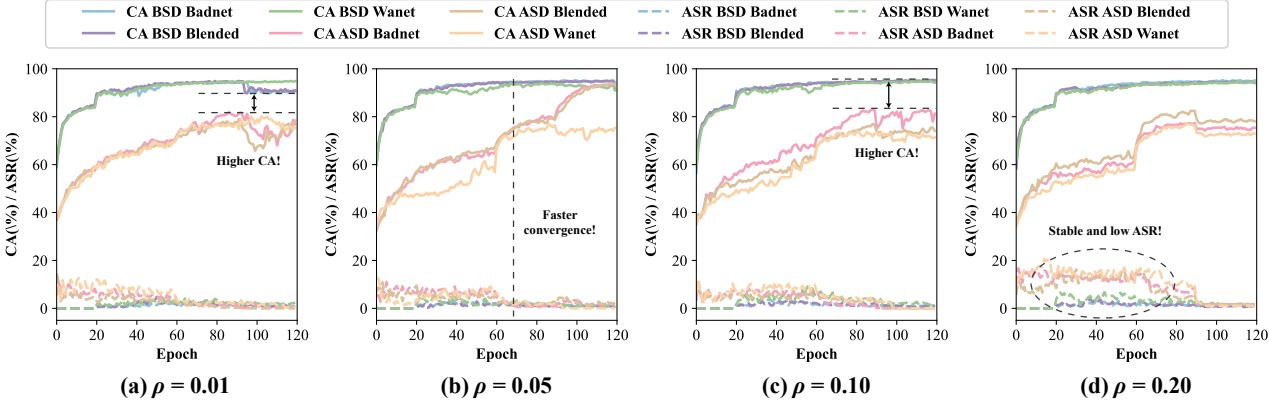

*Figure 3.* The performance of BSD in comparison with ASD (Gao et al., 2023) under different poisoning rates. The experiment is conducted on CIFAR-10 against three attacks.

*Table 3.* The clean accuracy (CA%), attack success rate (ASR%), and defense effectiveness rating (DER%) on CIFAR-10 of different defenses using mobilenet v2 (Sandler et al., 2018) as the backbone.

| Attack & Metric | | Non | FP | NAD | ABL | DBD | ASD | BSD |
|---|---|---|---|---|---|---|---|---|
| BadNet | CA | 94.3 | 77.9 | 78.5 | 79.7 | 65.5 | **93.2** | 91.1 |
| | ASR | 100.0 | 8.3 | 11.7 | 13.6 | **0.0** | 100.0 | 0.4 |
| | DER | - | 87.7 | 86.2 | 85.9 | 85.6 | 49.4 | **98.2** |
| Blended | CA | 94.0 | 75.9 | 76.0 | 67.3 | 69.0 | 87.1 | 90.0 |
| | ASR | 99.3 | 30.8 | 46.0 | 2.6 | **0.0** | 99.0 | 0.2 |
| | DER | - | 75.2 | 67.7 | 85.1 | 87.2 | 46.7 | **97.6** |
| WaNet | CA | 94.0 | 82.2 | 81.5 | 50.9 | 58.4 | 83.0 | 90.1 |
| | ASR | 95.7 | 2.4 | 3.2 | **0.5** | 12.4 | 97.7 | 0.6 |
| | DER | - | 90.7 | 90.0 | 76.1 | 73.9 | 44.5 | **95.6** |
| Average DER | | - | 84.5 | 81.3 | 82.3 | 82.2 | 46.9 | **97.1** |

### 5.4. Robustness to different poisoning rates

Despite the default poisoning rate $\rho = 0.05$ being a reasonable setting that is widely adopted in either backdoor attack or backdoor defense research (Huang et al., 2022; Gao et al., 2023; Min et al., 2024; Shi et al., 2023), it's crucial to verify the robustness of our BSD under different poisoning rates. As illustrated in Figure 3, although ASD performs well with respect to ASRs as well, the CAs of ASD are conspicuously lower under non-default settings. However, our BSD consistently achieves close-to-zero ASRs and satisfying CAs, emphasizing its robustness to different poisoning rates.

### 5.5. Robustness against different target labels

We evaluated the robustness against different targets of our BSD in Table 4. BSD presents consistently high performance against different target labels.

### 5.6. Training cost evaluation

Our BSD incorporates an altruistic model to assist with pool initialization and updates, which may raise concerns about increased training costs. However, as shown in Table 5, the training cost of BSD is comparable to, or even lower than,

*Table 4.* The clean accuracy (CA%), attack success rate (ASR%), and defense effective rating (DER%) of our BSD against 3 representitive backdoor attacks with different target labels on CIFAR-10.

| Target | BadNets | | | Blended | | | WaNet | | |
|---|---|---|---|---|---|---|---|---|---|
| | CA | ASR | DER | CA | ASR | DER | CA | ASR | DER |
| 0 | 95.0 | 0.8 | 99.6 | 95.0 | 0.4 | 99.0 | 91.9 | 0.3 | 99.0 |
| 1 | 94.9 | 0.5 | 99.8 | 94.9 | 0.5 | 98.9 | 94.2 | 0.3 | 99.8 |
| 2 | 95.1 | 0.8 | 99.6 | 94.7 | 0.9 | 98.7 | 90.9 | 0.7 | 98.3 |
| 3 | 95.1 | 0.9 | 99.6 | 94.9 | 0.8 | 98.8 | 94.5 | 0.3 | 99.6 |
| 4 | 95.0 | 0.2 | 99.9 | 94.8 | 0.6 | 98.9 | 92.4 | 0.2 | 99.3 |
| 5 | 95.1 | 1.7 | 99.2 | 95.0 | 0.5 | 98.9 | 91.9 | 0.4 | 98.9 |
| 6 | 95.1 | 0.6 | 99.7 | 93.9 | 0.6 | 98.8 | 92.6 | 0.3 | 99.3 |
| 7 | 94.7 | 0.3 | 99.8 | 92.6 | 0.4 | 98.2 | 90.3 | 0.0 | 98.3 |
| 8 | 92.0 | 0.3 | 98.4 | 95.1 | 0.5 | 98.9 | 91.8 | 0.3 | 98.9 |
| 9 | 94.9 | 0.3 | 99.8 | 94.0 | 0.4 | 98.9 | 94.0 | 0.2 | 99.9 |
| Avg | 94.7 | 0.6 | 99.5 | 94.5 | 0.6 | 98.8 | 92.5 | 0.4 | 99.1 |

that of ASD (Gao et al., 2023). This is due to three key factors: 1) The altruistic model is updated through standard training rather than MixMatch, significantly reducing time. 2) The altruistic model is only updated before stage 3, and its training primarily runs in parallel with the main model. 3) An imbalanced pool size, as seen in the early stages of ASD, often triggers frequent dataloader updates in MixMatch, whereas the clean pool size in BSD is more balanced and suitable during training.

*Table 5.* Training cost (hours) of ASD, DBD, and BSD on CIFAR-10, GTSRB, and Imagenet.

| Method | CIFAR-10 | GTSRB | Imagenet | Average |
|---|---|---|---|---|
| DBD | 11.96 | 10.09 | 53.21 | 25.09 |
| ASD | 4.81 | 2.55 | 12.09 | 6.48 |
| BSD(ours) | 3.15 | 2.84 | 9.20 | 5.06 |

### 5.7. Ablation studies

**Effectiveness of different stages.** The major components of BSD are divided into pool initialization and pool updates. We investigated the significance of each component on CIFAR-10 to demonstrate their necessity, as shown in Table 6. OSS and ALS initialization are critical for avoid-

ing backdoor overfitting (ASR); class completion update helps prevent class underfitting (CA); and selective dropping update acts as a final step to further reduce ASR, thereby achieving a higher DER.

*Table 6.* The ablation study on the strategies involved in BSD under CIFAR-10. 'Default' represents the result using all the proposed mechanisms, 'w/o Init' represents the results using random initialization. 'w/o Comp' represents disabling class completion in both stages 2 and 3. 'w/o Drop' represents disabling selective drop in stage 3.

| SETTING↓ | BADNET | | | BLENDED | | | WANET | | |
|---|---|---|---|---|---|---|---|---|---|
| | CA | ASR | DER | CA | ASR | DER | CA | ASR | DER |
| DEFAULT | 95.1 | 0.9 | 99.6 | 94.9 | 0.8 | 98.8 | 94.5 | 0.8 | 99.6 |
| W/O INIT | 94.7 | 100.0 | 49.9 | 95.0 | 99.2 | 49.6 | 93.6 | 91.5 | 54.2 |
| W/O COMP | 90.7 | 0.0 | 97.9 | 86.8 | 0 | 95.5 | 89.8 | 0.2 | 98.0 |
| W/O DROP | 94.6 | 1.1 | 99.3 | 94.2 | 1.1 | 98.6 | 94.5 | 1.9 | 99.0 |

**Influence of parameters.** We here present the influence of the main parameter, i.e., the parameters $\alpha$ and $\beta$ controlling the pool size. As revealed in Table 7, our BSD has robust performance against all the attacks with a relatively loose range of $\alpha$ and $\beta$ , and we recommend using the default setting in normal cases, and a reasonable range for adjustments is $0.3 < \alpha < 0.8, \beta < 0.5$.

*Table 7.* Performance of BSD under different $\alpha$ & $\beta$ on CIFAR-10. The results that have more than 0.5% DER decrease are marked using ↓.

| SETTING↓ | BADNET | | | BLENDED | | | WANET | | |
|---|---|---|---|---|---|---|---|---|---|
| | CA | ASR | DER | CA | ASR | DER | CA | ASR | DER |
| DEFAULT | 95.1 | 0.9 | 99.6 | 94.9 | 0.8 | 98.8 | 94.5 | 0.8 | 99.6 |
| $\alpha=0.3, \beta=0.2$ | 94.9 | 1.2 | 99.4 | 94.8 | 0.6 | 98.9 | 94.2 | 1.8 | 99.1↓ |
| $\alpha=0.4, \beta=0.2$ | 95.0 | 1.2 | 99.4 | 94.7 | 0.9 | 98.7 | 94.4 | 1.4 | 99.3 |
| $\alpha=0.5, \beta=0.2$ | 95.0 | 1.7 | 99.2 | 94.2 | 0.8 | 98.7 | 94.7 | 1.4 | 99.2 |
| $\alpha=0.7, \beta=0.2$ | 95.2 | 0.8 | 99.6 | 95.1 | 0.5 | 98.9 | 93.0 | 0.1 | 99.6 |
| $\alpha=0.8, \beta=0.2$ | 95.0 | 1.1 | 99.5 | 92.8 | 0.5 | 98.2↓ | 91.7 | 1.0 | 98.5↓ |
| $\alpha=0.9, \beta=0.2$ | 93.6 | 0.7 | 99.0↓ | 90.3 | 0.2 | 97.2↓ | 90.0 | 0.9 | 97.7↓ |
| $\alpha=0.6, \beta=0.1$ | 94.8 | 0.7 | 99.6 | 90.9 | 0.5 | 97.3↓ | 91.6 | 0.6 | 98.7↓ |
| $\alpha=0.6, \beta=0.3$ | 94.9 | 1.2 | 99.4 | 94.7 | 0.8 | 98.8 | 94.5 | 0.8 | 99.6 |
| $\alpha=0.6, \beta=0.5$ | 94.7 | 1.4 | 99.2 | 94.9 | 1.6 | 98.4 | 94.3 | 2.3 | 98.8↓ |
| $\alpha=0.7, \beta=0.7$ | 94.4 | 1.7 | 98.9↓ | 94.3 | 3.7 | 97.3↓ | 94.0 | 32.8 | 83.6↓ |

### 5.8. Extended experiments

Additional experimental results, including visualizations, extended ablation studies, more baselines, performance under no attacks, robustness of pseudo target approximation, resistance to all2all attacks, potential adaptive attacks, and more, are provided in Appendix D.

## 6. Conclusion

In conclusion, our proposed BSD effectively mitigates backdoor attacks through bi-perspective splitting mechanisms, without relying on on extra clean data.

## Acknowledgements

This work was supported by National Natural Science Foundation of China (Grant Numbers 61906040, 61972085, 62276063, 62272101, 6509009710), the National Key Research and Development Program of China (Grant Number 2022YFF0712400), the Natural Science Foundation of Jiangsu Province (Grant Number BK20221457, BK20230083), the Fundamental Research Funds for the Central Universities (Grant Number 2242021R41177), and SIP Support-startup funding (Grant Number MSRI8001004), and the Southeast University Interdisciplinary Research Program for Young Scholars.

## Impact Statement

This paper presents work whose goal is to advance the field of Machine Learning. There are many potential societal consequences of our work, none which we feel must be specifically highlighted here.

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

# A. Algorithm outline

The pseudocode of the proposed method BSD is listed as Algorithm 1.

---

**Algorithm 1** Pseudocode for BSD

---

**Input:** Poisoned training set $\mathcal{D}$; main model $f$; main model warm-up ends at epoch $T_1$, main model training stage2 ends at epoch $T_2$, main model training stage3 ends at epoch $T_3$, max clean pool ratio $\alpha$, OSS split ratio $\beta$.

**Output:** Clean model $f_{\theta'}$

1: **# Initialization & warm-up**
2: Initialize the weights of $f$ as $\theta$
3: Generate an altruistic model $g$ having the same architecture as $f$, initialize the weights as $\varphi$
4: # Prepare for ALS
5: **for** i = 1 to 25 **do**
6:    **for** each sample $(x, y)$ in $\mathcal{D}$ **do**
7:       $loss \leftarrow \mathcal{L}_{ce}(x, y, g_{\varphi})$
8:       $\varphi \leftarrow \varphi - \nabla_{\varphi} loss$
9:    **end for**
10: **end for**
11: # Prepare for OSS
12: Set $y_t$ as the majority of most frequent second likely prediction, i.e., $y_t = \text{Majority}(\text{argsort}(-logit_i)[1])$.
13: Calculate $\mathcal{D}_t$ and $\mathcal{D}_{nt}$ with $y_t$ according to Section 3.2
14: **# Main Training Loop**
15: **while** $T < T_3$ **do**
16:    **if** $T < T_1$ **then**
17:       # Data used for the main model warm-up
18:       $\mathcal{D}_c \leftarrow \mathcal{D}_{nt}$
19:    **else if** $T = T_1$ **then**
20:       # Pool initialization using ALS and OSS
21:       $\mathcal{D}_c \leftarrow \mathcal{D}_{als} \cup \mathcal{D}_{oss}$
22:    **else if** $T_1 + 10 \leq T < T_2$ **then**
23:       # Pool update based on loss discrepancy of $f_{\theta}$ and $g_{\varphi}$, enabling class completion
24:       $T' \leftarrow \frac{T - T_1 - 10}{T_2 - T_1 - 10} T_2$
25:       Current clean ratio $\alpha_T \leftarrow \beta + (\alpha - \beta) \times (1 - \cos(\pi \times T'/T_2))/2$
26:       Set $\alpha$ as $\alpha_T$ in (8)
27:       Calculate $\mathcal{D}_{\tilde{c}_1}$ according to (8)
28:       $\mathcal{D}_c \leftarrow \mathcal{D}_{\tilde{c}_1}$
29:    **else if** $T \geq T_2$ **then**
30:       # Pool update based on loss discrepancy of $f_{\theta}$ and $g_{\varphi}$, enabling class completion and selective drop
31:       Current clean ratio $\alpha_T \leftarrow \alpha$
32:       Set $\alpha$ as $\alpha_T$ in (10)
33:       Calculate $\mathcal{D}_{\tilde{c}_2}$ according to (10)
34:       $\mathcal{D}_c \leftarrow \mathcal{D}_{\tilde{c}_2}$
35:    **end if**
36:    $\mathcal{D}_p \leftarrow \mathcal{D} \setminus \mathcal{D}_c$
37:    # Models updating
38:    $\theta \leftarrow \theta - \nabla_{\theta} \mathcal{L}_{semi}$ # Train the model on $\mathcal{D}_c$(labeled) and $\mathcal{D}_p$ by semi-supervised learning
39:    **if** $T < T_2$ **then**
40:       $\varphi \leftarrow \varphi - \nabla_{\varphi} \mathcal{L}_{ce}$ # Train the altruistic model by supervised learning
41:    **end if**
42: **end while**

---

# B. Implementation details

## B.1. Environments

We run all the experiments using PyTorch on a Linux server with an AMD EPYC 7H12 64-core Processor, 256GB RAM, and $8\times$ NVIDIA GeForce RTX 3090 GPU.

## B.2. Illustration of the poisoned samples

Figure 4 illustrates the seven attack types used in this study, displaying both the original and poisoned images along with the corresponding trigger patterns. For attacks involving a different trigger in the Imagenet dataset, the specific trigger is also shown at the bottom.

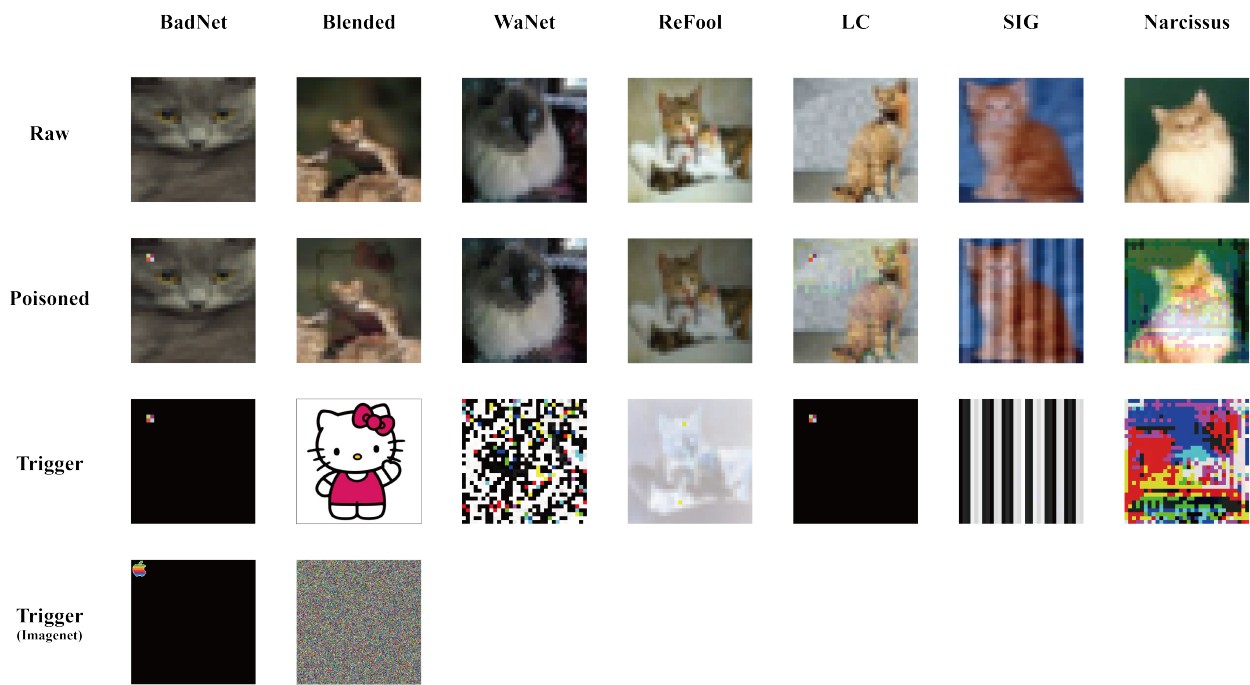

*Figure 4.* Illustation of the backdoor attacks. We present the examples on CIFAR-10, alternative triggers (if used) on Imagenet are shown at the bottom.

## B.3. Attack settings

**Training settings.** For all the attack implementations, we follow that in ASD (Gao et al., 2023). On the CIFAR-10 and GTSRB datasets, we perform backdoor attacks on ResNet-18 for 200 epochs with batch size 128. We adopt the stochastic gradient descent (SGD) optimizer with a learning rate of 0.1, momentum of 0.9, and weight decay $5 \times 10^{-4}$. The learning rate is divided by 10 at epoch 100 and 150. For attacks not achieving reported performance in ASD (Gao et al., 2023), we continue the training for another 100 epochs, and the learning rate is divided by 10 at epoch 200 and 250. On the Imagenet (Deng et al., 2009) dataset, we train ResNet-18 for 90 epochs with batch size 256. We utilize the SGD optimizer with a learning rate of 0.1, momentum of 0.9, and weight decay $1 \times 10^{-4}$. The learning rate is decreased by a factor of 10 at epoch 30 and 60. The image resolution will be resized to $224 \times 224 \times 3$ before attaching the trigger pattern.

**Settings for BadNets.** As suggested by (Gu et al., 2017; Huang et al., 2022; Gao et al., 2023), we set a $2 \times 2$ square on the upper left corner as the trigger pattern on CIFAR-10 and GTSRB. For ImageNet and VGGFace2, we use a 32×32 apple logo on the upper left corner. The poisoning rate $\rho$ is set to 0.05(5%).

**Settings for Blended.** Following (Chen et al., 2017; Huang et al., 2022; Gao et al., 2023), we choose"Hello Kitty" pattern on CIFAR-10 and GTSRB and the random noise pattern on ImageNet and VGGFace2. The blend ratio is set to 0.1. The poisoning rate $\rho$ is set to 0.05(5%).

**Settings for WaNet.** Following (Gao et al., 2023; Huang et al., 2022), we directly use the default warping-based operation to generate the trigger pattern. For CIFAR-10 and GTSRB, we set the noise rate $\rho_n$ to 0.2, control gird size $k$ as 4, and warping strength $s$ as 0.5. For Imagenet, we use the same noise rate, but a larger grid size $k = 224$, and a warping strength $s = 1$.

**Settings for Label-Consistent Attack.** Following (Gao et al., 2023; Huang et al., 2022; Turner et al., 2019), the noisy versions of samples are generated using adversarially trained models. The PGD parameters are as follows: for PGD training: $\epsilon = 16$, $\alpha = 2$, steps = 7, and the pixel range is [0, 255]; for PGD attack: $\epsilon = 16$, $\alpha = 1.5$, steps = 30, with the same pixel range [0, 255]. The same trigger used in BadNets is applied for LC attacks, and the poison ratio is set at 25

**Settings for Refool.** Following (Li et al., 2021a; Liu et al., 2020), we randomly choose 5,000 images from PascalVOC (Everingham et al., 2015) as the candidate reflection set $\mathcal{R}_{cand}$ and randomly choose one of the three reflection methods to generate the trigger pattern during the backdoor attack.

**Settings for SIG.** Following (Li et al., 2021a; Barni et al., 2019), we adopt the same sinusoidal pattern in ABL as the trigger and set the poisoning rate to match LC, as SIG is a clean-label attack.

**Settings for Narcissus.** We also incorporate the recent attack proposed by (Zeng et al., 2023b), which is another clean-label attack. The parameter settings for generating the Narcissus trigger pattern are as follows: the $\ell_\infty$ ball bound is set to 16/255, the surrogate model is trained for 200 epochs with an initial learning rate of 0.1 and a warm-up period of 5 rounds. The trigger-generation learning rate is 0.01, and the generation process lasts for 1000 rounds. The poisoning rate is the same as LC, given that Narcissus is also a clean-label attack.

### B.4. Defense settings

**Settings for FP.** Following (Gao et al., 2023), we set two steps of FP (Liu et al., 2018) (i.e., pruning and fine-tuning) as follows. (1) We randomly select 5% clean training samples as the local clean samples and forward them to obtain the activation values of neurons in the last convolutional layer. The dormant neurons on clean samples with the lowest $\alpha\%$ activation values will be pruned. (2) The pruned model will be fine-tuned on the local clean samples for 10 epochs. In particular, the learning rate is set as 0.01, 0.01, 0.1 on CIFAR-10, GTSRB, and ImageNet. Unless otherwise specified, other settings are the same as those used by (Liu et al., 2018). For the hyper-parameters of FP, we search for the best results by adjusting the pruned ratio $\alpha\% \in 20\%, 30\%, 40\%, 50\%, 60\%, 70\%, 80\%, 90\%$. In addition, we add another default setting in backdoorbench (Wu et al., 2022).

**Settings for NAD.** NAD (Li et al., 2021b) is also a post trianing method that repairs the backdoored model and needs 5% local clean training samples. We set the two steps of NAD as follows: (1) Use the local clean samples to fine-tune the backdoored model for 10 epochs. Specially, the learning rate is set as 0.01, 0.01, 0.1 on CIFAR-10, GTSRB, and ImageNet. (2) The fine-tuned model and the backdoored model will be regarded as the teacher model and student model to perform the distillation process. Unless otherwise specified, other settings are the same as those used by (Li et al., 2021b). For the sensitive hyper-parameter $\beta$, we find the search space used by (Gao et al., 2023) too small. We search for the best results by adjusting the hyper-parameter $\beta$ from 500, 1000, 1500, 2000, 2500, 5000, 7500, 1e5, 1e6, 1e7, 1e8, 1e9, 1e10, 1e11. In addition, we add another default setting in backdoorbench (Wu et al., 2022).

**Settings for ABL.** ABL (Li et al., 2021a) contains three stages: (1) To obtain the poisoned samples, ABL first trains the model on the poisoned dataset for 20 epochs by LGA loss and isolate 1% training samples with the lowest loss. (2) Continue to train the model with the poisoned dataset after the backdoor isolation for 70 epochs. (3) Finally, the model will be unlearned by the isolation samples for 5 epochs. The learning rate is 5e-4 at the unlearning stage. Unless otherwise specified, other settings are the same as those used by (Li et al., 2021a). ABL is sensitive to the hyper-parameter $\gamma$ in LGA loss. We search for the best results by adjusting the hyper-parameter $\gamma$ from 0, 0.1, 0.2, 0.3, 0.4, 0.5, In addition, we add another default setting in backdoorbench (Wu et al., 2022).

**Settings for DBD.** DBD (Huang et al., 2022) contains three independent stages: (1) DBD uses SimCLR to perform the self-supervised learning for 1,000 epochs. (2) Freeze the backbone and fine-tune the linear layer by supervised learning for 10 epochs. (3) Adopt the MixMatch to conduct the semi-supervised learning for 200 epochs on CIFAR-10 and GTSRB for 90 epochs on ImageNet and VGGFace2. Unless otherwise specified, other settings are the same as those used by (Huang et al., 2022). Since DBD is a relatively stable backdoor defense and not sensitive to its hyper-parameter, we only add another group of default setting in backdoorbench (Wu et al., 2022).

**Settings for ASD.** We follow the exact settings for ASD as suggested by (Gao et al., 2023). To name a few settings, we adopt MixMatch as the semi-supervised learning framework and use the Adam optimizer with a learning rate of 0.002 and a batch size of 64 for the semi-supervised training. The temperature $T$ is set to 0.5, and the weight of the unsupervised loss $\lambda_u$ is set to 15. The training stages are defined as follows: $T_1 = 60$, $T_2 = 90$, and $T_3 = 120$ for CIFAR-10 and ImageNet, while $T_3 = 100$ for GTSRB. Similarly, other parameters are the same as used by (Gao et al., 2023) as well.

For our BSD, we adopt the MixMatch (Berthelot et al., 2019b) semi-supervised training framework for the main model, following Decoupling-based Defense (DBD) and Adaptive Splitting-based Defense (ASD). The semi-supervised learning parameters align with ASD, including 1024 training iterations, a temperature of 0.5, a ramp-up length of 120, and a learning rate of 0.002. The altruistic model undergoes a warm-up phase with 25 epochs, utilizing the Adam optimizer, Cross Entropy loss, with a learning rate of 0.001. The default warm-up epochs for the main model in OSS are set to 20 (followed by a 10-epoch training on the initialized pools)($T_1 = 20$), with a default $\beta$ of 0.2. Class completion training spans 60 epochs ($T_2 = 90$), and selective dropping training spans 30 epochs ($T_3 = 120$). The altruistic model update uses the same loss and optimizer as in the warm-up on CIFAR-10 and Imagenet for efficiency, on lightweight datasets like GTSRB, we use the same semi-supervised loss and optimizer as the main model for better performance. The clean pool ratio $\alpha$ follows a sinusoidal growth curve during class completion training, starts at $\beta$, and reaches an upper limit of $\alpha = 0.6$ at the end of the class completion stage, after which it remains fixed:

$$T' = \frac{T - T_1 - 10}{T_2 - T_1 - 10} T_2$$
$$\alpha_T = \beta + (\alpha - \beta) \times (1 - \cos(\pi \times T'/T_2))/2 \tag{11}$$

The baselines are implemented using:

- BackdoorBench (Wu et al., 2022);

- BackdoorBox (Li et al., 2023b);

- Github repositories of corresponding papers.

We greatly appreciate these outstanding works.

## B.5. Definition of DER

Defense Effectiveness Rating (DER) (Zhu et al., 2023a) is a comprehensive measure that considers both ACC and ASR:

$$\text{DER} = [\max(0, \Delta\text{ASR}) - \max(0, \Delta\text{ACC}) + 1]/2, \tag{12}$$

where $\Delta\text{ASR}$ denotes the decrease of ASR after applying defense, and $\Delta\text{ACC}$ denotes the drop in ACC following the defense. Higher ACC, lower ASR and higher DER indicate better defense performance.

## B.6. Details about semi-supervised loss

Semi-supervised learning (Berthelot et al., 2019a;b; Sohn et al., 2020; Xie et al., 2020; Zhu & Goldberg, 2022) studies how to leverage a training dataset with both labeled data and unlabeled data to obtain a model with high accuracy. In addition to its application in normal training, semi-supervised learning also serves as a powerful means for the security of DNNs (Alayrac et al., 2019; Carmon et al., 2019; Huang et al., 2022).

Here we adopt the MixMatch (Berthelot et al., 2019b). Given a batch $\mathcal{X} \subset \mathcal{D}_C$ of labeled samples, and a batch $\mathcal{U} \subset \mathcal{D}_P$ of unlabeled samples, MixMatch generates a guessed label distribution $\tilde{q}$ for each unlabeled sample $u \in \mathcal{U}$ and adopts MixUp to augment $\mathcal{X}$ and $\mathcal{U}$ to $\mathbf{X}\prime$ and $\mathbf{U}\prime$. The supervised loss $\mathcal{L}_s$ is defined as:

$$\mathcal{L}_s = \sum_{(x,q) \in \mathcal{X}'} \text{H}(p_x, q), \tag{13}$$

where $p_x$ is the prediction of $x$, $q$ is the one-hot label and $H(\cdot, \cdot)$ is the cross-entropy loss. The unsupervised loss $\mathcal{L}_u$ is defined as:

$$\mathcal{L}_u = \sum_{(u,\bar{q}) \in \mathcal{U}'} \|p_u - \bar{q}\|_2^2, \tag{14}$$

where $p_u$ is the prediction of $u$.

Finally, the MixMatch loss can be defined as:

$$\mathcal{L} = \mathcal{L}_s + \lambda \cdot \mathcal{L}_u, \tag{15}$$

where $\lambda$ is a hyper-parameter for trade-off, we adopt the same $\lambda = 15$ as in ASD.

## C. Supplementary information of the background

### C.1. Supplementary Overview of Backdoor Attack Research

The common implementation of backdoor attacks is realized by injecting a few poisoned samples into the training dataset, i.e., data-poisoning-based backdoor attacks, inducing the model to build a link between the trigger (i.e., a visual particular pattern) and target class (Gu et al., 2017). Thus the model consistently outputs the target label once the trigger is attached to the inputs in the inference stage.

Poison-label backdoor attacks are currently the most common attack paradigm, where the trigger pattern in the poisoned samples is directly connected to the target class by relabeling, inducing the model to treat the trigger as a decision-making feature of the target class. Recent research (Hu et al., 2022; Li et al., 2020; Qi et al., 2021) focuses on more invisible trigger designs through generative models and feature space optimizations, as well as exploring backdoor attacks in wider tasks like natural language processing.

### C.2. Extended related works

With the advance of clean subset extraction and backdoor detection, many works tries to split clean subsets from poison training sets. (Zeng et al., 2023a) proposed detecting poisoned data by identifying shifts in data distributions, which results in high prediction loss when training on the clean portion of a poisoned dataset and testing on the corrupted portion. They solve a relaxed of the splitting optimization problem with the help of a weight-assigning network. Although promising empirical results were presented, the proposed META-SIFT only guarantees a relatively small subset ((Zeng et al., 2023a), page 10, Figure 5). As a result, META-SIFT still relies on effective downstream defenses, such as NAD and ASD, included in our baselines, while also increasing the hyperparameter search space. (Pan et al., 2023) are motivated by the same distributional shift phenomenon and proposed an effective splitting algorithm, ASSET. However, they assume that the defender has an extra set of clean samples (named "base set" in (Pan et al., 2023)), which doesn't suit the background of our paper, where no extra clean set is available. Plus, ASSET is faced with the same problem that requires effective downstream defenses to conduct the defense.

In general, these works indeed provide valuable insights into the poisoned data splitting problem and could inspire our future research. However, they are faced with two major problems. 1) Cannot guarantee a 100% correct split that can be directly used for training; 2) Rely on an extra clean set which violates the constraints of our scenario.

### C.3. Illustration of the model collapse

As presented in Figure 5, the splitting-based defenses (loss-guided ones specifically) encounter two kinds of model collapse. In backdoor overfitting collapse, poison samples take effect and have low loss values, which consistently corrupt the clean pool and lead to a backdoored model. Likewise, in class underfitting collapse, the rareness of certain classes will lead to higher loss values, making them less chosen to be clean samples, which forms a vicious cycle. Note that these two collapses are common in other categories of defenses as well, while it's more explainable in splitting-based defense.

## D. Extended experimental results

### D.1. Illustration of pool update

To showcase the healthy clean pool acquired by our BSD, we plot the number of poison samples in the clean pool at each training epoch, as well as reveal the accumulated number of poison samples. As shown in Figure 6, our BSD generally have fewer poison samples in the clean pool during training, with both the number of poisoned samples and the cumulative number of samples smaller than that fo ASD under different poisoning rates.

In addition, we plot the loss/distance distribution of samples of our BSD in Figure 7. In Figure 7.(a) and Figure 7.(b),

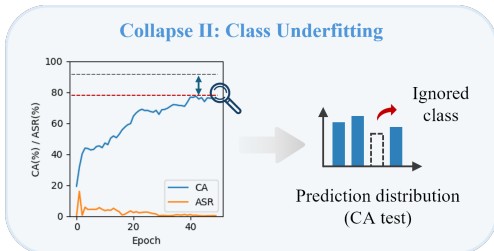

*Figure 5.* Typical model collapses in data-splitting backdoor defenses. **I**: Misclassification of low-loss poisoned samples as clean leads to a steady increase in the poisoned sample proportion until 100% ASR. **II**: Higher losses for challenging classes reduce their presence in the clean pool, rendering the model unable to predict samples from those categories.

the main mechanisms, i.e., ALS and OSS for the pool initialization, successfully distinguished the poison samples. In Figure 7.(c), the poison samples are highlighted by the high loss discrepancy between the main model and the altruistic model. The final result shown in Figure 7.(d) reveals the high CAs (clean sample all having low loss values) and low ASRs (poison samples all having high loss values) of BSD.

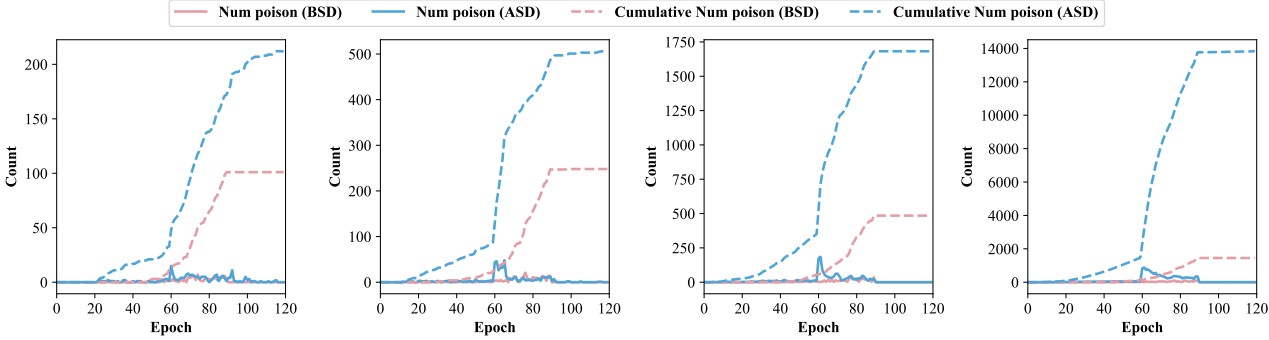

*Figure 6.* The number of poison samples in the clean pool of BSD and ASD at each epoch, the accumulated number in the dotted line. The subplots are the results on CIFAR-10, $\rho = 0.01, 0.05, 0.10, 0.20$.

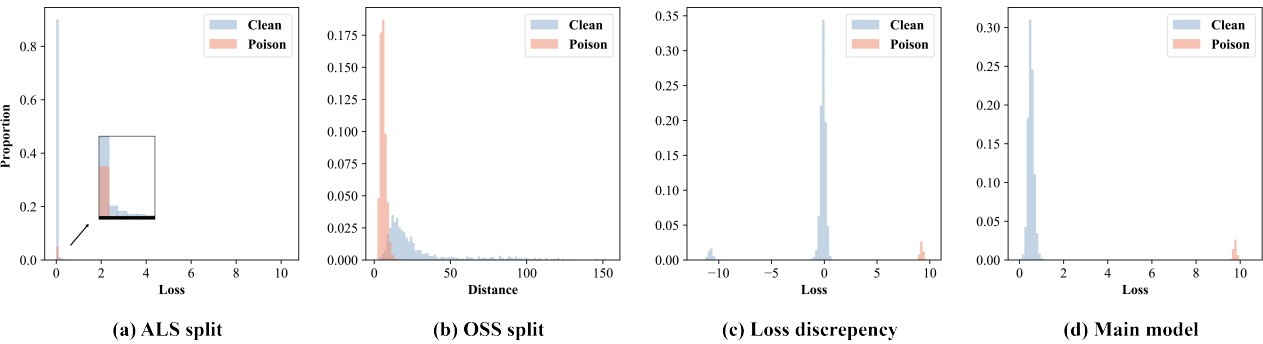

*Figure 7.* The split visualization of BSD on GTSRB against the BadNet attack. (a) the loss distribution on the altruistic model after the ALS warm-up; (b) the distance distribution on the main model after the OSS warm-up; (c) the loss discrepancy at the last epoch; (d) the loss distribution on the main model at the last epoch.

### D.2. Influence of different settings in OSS

**Ablation on distance metric of OSS.** We investigate the influence of the number of runs for the warm-up and three different distance calculations of OSS as shown in Figure 8. For the distance calculation, we take three approaches, i.e., the minimal,

the maximal, and the mean $\ell_2$ distance to feature clusters of each non-target category. Intuitively, we set the minimal distance by default, because the poisoned samples we consider are characterized by being far away from all existing cluster centers, thus maximal and mean distances may misjudgment two categories whose original clustering centers are far apart from each other as poisoned samples. Whereas, all three approaches exhibit good separation under the default 20-epoch warm-up.

**Ablation on warm-up epochs of OSS** ($T_1$)**.** Concerning the number of warm-up epochs, we investigate the score distribution of OSS under the min-distance calculation. As illustrated in Figure 8, the result exhibits poor separation with an insufficient warm-up. As the number of epochs goes up, it has a certain effect when the number of epochs equals 10, and perfectly separates some benign samples with larger epochs.

Figure 8 is the complete result of the ablation study on different settings of the warm-up process of OSS. It verifies the effectiveness of all three distance metrics. Intuitively, the model should have a more separable and reliable initialization of the two pools with a long warm-up, whereas the result of a 40-epoch warm-up (especially when using the mean distance) violates this intuition by exhibiting less satisfying separation. A potential reason is that the model overfitted the non-target classes, thus the poison samples have less similarity to them.

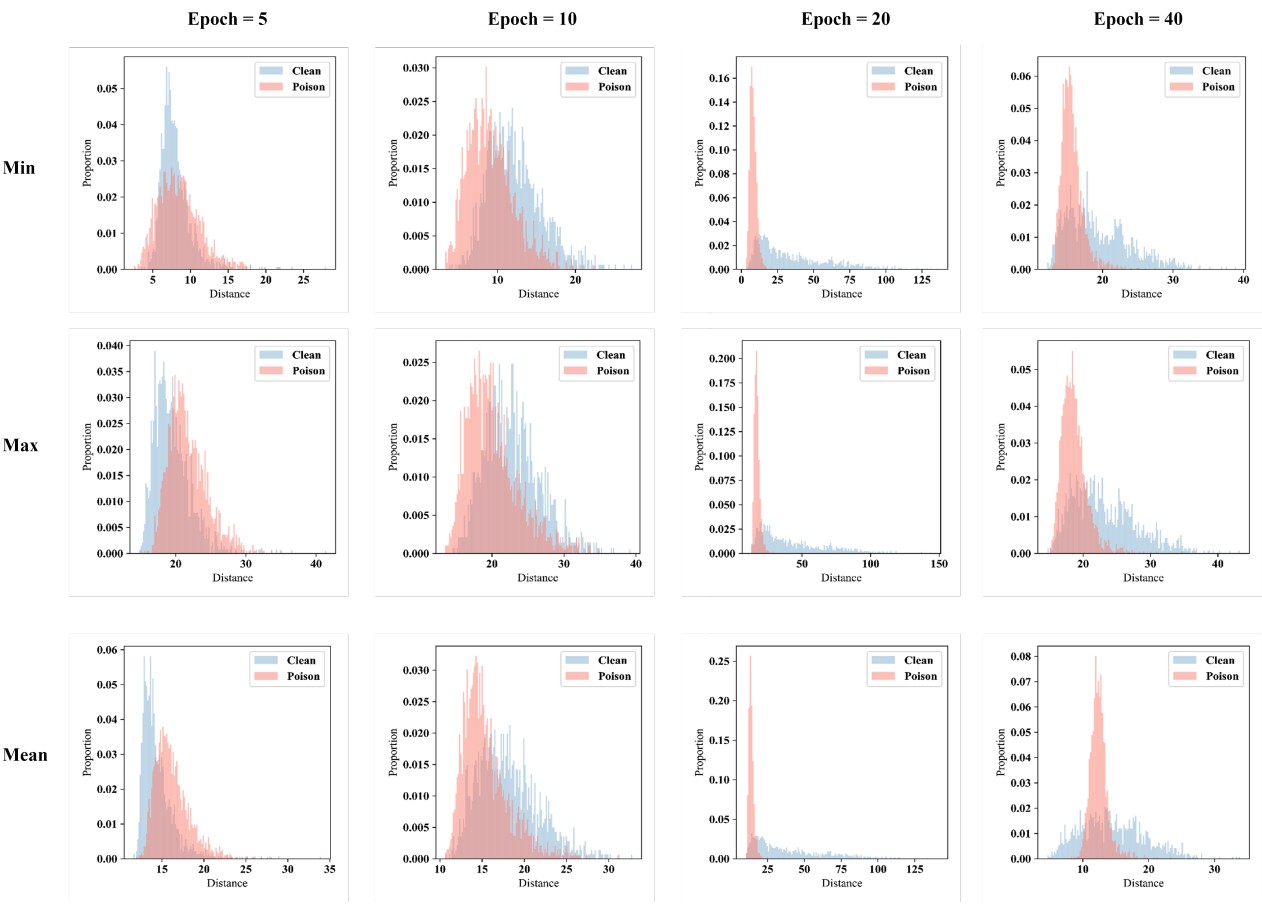

*Figure 8.* The ablation results on the number of warm-up epochs and different distance calculation methods for OSS.

In general, the default setting of BSD is a suitable choice.

### D.3. Influence of different settings in ALS

**Ablation study on warm-up epochs of the altruistic model** As our $y_t$ estimation method takes the majority within the warm-up epochs in ALS, we visualize the prediction in each single epoch in Figure 9. As seen in this figure, all the final majority predictions and the most internal majority prediction of $y_t$ are the same ($y_t = 3$), which is the ground truth target

label.

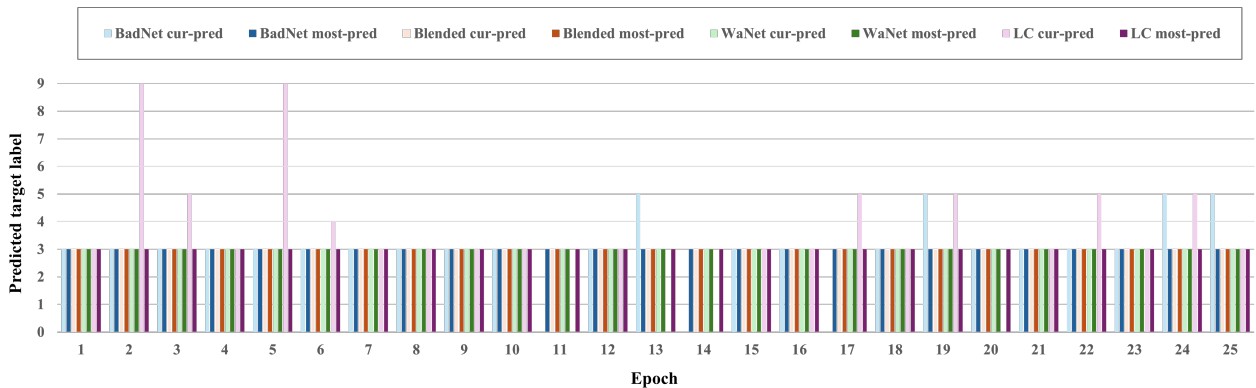

*Figure 9.* The result of the alternative method for $y_t$ approximation. In the default 25 epochs of warm-up, we count the pred $y_t$ at each epoch and the most pred $y_t$ by that epoch respectively. The experiment is conducted on CIFAR-10, against BadNet, Blended, WaNat, and LC.

**Alternative method for approximating** $y_t$  For unseen failures that we failed to correctly approximate $y_t$, we provide an alternative method for approximating $y_t$ against new backdoor attacks that may appear in the future. We here adopt a lightweight solution by just slightly modifying the warm-up of the altruistic model. We add local gradient ascent (Li et al., 2021a) and a local voting process: $\tilde{y}_t = \arg\max_c |\{(x, y) \mid y = c \wedge (x, y) \in \mathcal{D}_{lga}\}|$, where $\mathcal{D}_{lga}$ denotes the isolated 1% samples having the smallest loss values after local gradient ascent training on the altruistic model. In common scenarios where the dataset is a large but well-known benchmark dataset, the number of samples in each class is known to the public, $y_t$ can be just approximated through label statistics.

Notably, for this alternative $y_t$ estimation method, if the detection precision exceeds 50%, it indicates that more than half of the isolated samples are poison samples, thus we can obtain $y_t$. Although the experimental results presented by (Li et al., 2021a) in their Figure 7, page 16 has already verified a more than 50% against most common attacks, we further check its robustness to the warm-up epochs in Table 8.

*Table 8.* The prediction of $y_t$ under different warm-up epochs. The 'num' represents the number of poison samples in the isolated set.

| WARM-UP EPOCHS | BADNET | | BLENDED | |
|---|---|---|---|---|
| | NUM POISON | $y\_t$ | NUM POISON | $y\_t$ |
| 5 | 0 | WRONG | 0 | WRONG |
| 15 | 461 | CORRECT | 457 | CORRECT |
| 25 | 370 | CORRECT | 344 | CORRECT |
| 35 | 85 | CORRECT | 126 | WRONG |
| 45 | 177 | CORRECT | 114 | CORRECT |

### D.4. Additional baselines

We added two recent defense, VaB (Zhu et al., 2023b) and D-ST/D-BR (Chen et al., 2022b). The additional baselines are implemented based on the official implementation. We use CIFAR-10 as the dataset. Since the label-consistent attack is not consistently implemented, we use SIG as a clean label attack here. As shown in Table 9, VaB has the most competitive result against poison label attacks, but struggles to defend against SIG.

### D.5. Performance under no attacks

Most backdoor defense research focuses on performance under attack, while it is concerning that these defenses may degrade model performance in the absence of attacks. Therefore, we evaluated the performance of BSD in scenarios without attacks. There are no poisoned samples in the training set, we test the clean and poisoned samples (BadNets trigger) for

*Table 9.* The clean accuracy (CA%), attack success rate (ASR%), defense effective rating (DER%) and time cost (hours) of 2 additional backdoor defense methods and our BSD against 4 threatening backdoor attacks on CIFAR-10. The best and second best results are in **bold** and underlined.

| METHOD | BADNETS | | | BLENDED | | | WANET | | | SIG | | | AVG DER | TIME COST |
|---|---|---|---|---|---|---|---|---|---|---|---|---|---|---|
| | CA | ASR | DER | CA | ASR | DER | CA | ASR | DER | CA | ASR | DER | | |
| VAB | 94.0 | 1.3 | 98.9 | 94.2 | 1.1 | 98.6 | 93.6 | 1.7 | 99.1 | 94.0 | 66.6 | 63.8 | 90.1 | 5.5 |
| D-ST | 66.8 | 5.7 | 83.1 | 65.0 | 7.1 | 81.1 | 60.8 | 15.2 | 76.0 | 87.9 | 95.1 | 46.5 | 71.6 | 4.3 |
| D-BR | 87.5 | **0.8** | 95.9 | 83.0 | 80.7 | 53.2 | 16.9 | 14.6 | 54.3 | 85.7 | 0.1 | 92.9 | 74.1 | - |
| BSD(OURS) | **95.1** | 0.9 | **99.6** | **94.9** | **0.8** | **98.8** | **94.5** | **0.8** | **99.6** | 93.8 | **0.0** | **97.0** | **98.7** | **3.2** |

inference. As shown in Table 10, it is worth noting that even in the absence of attacks, there can be a low attack success rate (ASR), where these samples are just being misclassified to the target label. As Table 10 reveals, our method achieves a lower ASR compared to the baseline, effectively suppressing natural backdoors. Additionally, our method shows significant improvements in accuracy over the baseline.

*Table 10.* The clean accuracy (CA%) and attack success rate (ASR%) of BSD and ASD under no attacks.

| METHOD | CIFAR-10 | | GTSRB | |
|---|---|---|---|---|
| | ACC | ASR | ACC | ASR |
| NO DEFENSE | 95.3 | 1.9 | 97.7 | 0.2 |
| ASD | 93.2 | 1.8 | 96.6 | 0.1 |
| BSD (OURS) | 94.9 | 0.6 | 97.6 | 0.0 |

### D.6. Robustness of pseudo target approximation

**Pseudo target approximation test on GTSRB.** We evaluated the approximation of $y_t$ on the GTSRB dataset with various ground truth target labels, as shown in Table 11 (using the main approximation method). The results demonstrate that $y_t$ was successfully approximated for all of the first 10 classes in GTSRB.

*Table 11.* Testing the $y_t$ approximation on different target labels (the first 10 classes) on GTSRB.

| ATTACK | 0 | 1 | 2 | 3 | 4 | 5 | 6 | 7 | 8 | 9 |
|---|---|---|---|---|---|---|---|---|---|---|
| BADNETS | ✓ | ✓ | ✓ | ✓ | ✓ | ✓ | ✓ | ✓ | ✓ | ✓ |
| BLENDED | ✓ | ✓ | ✓ | ✓ | ✓ | ✓ | ✓ | ✓ | ✓ | ✓ |
| LC | ✓ | ✓ | ✓ | ✓ | ✓ | ✓ | ✓ | ✓ | ✓ | ✓ |

**Performance under forced incorrect pseudo target label.** We conducted interesting additional tests by forcing $y_t$ to be assigned to an incorrect class and observed the model's performance (on CIFAR-10, against BadNets). As illustrated in Figure 10, BSD retained partial defensive capabilities even when the pseudo-label was deliberately set incorrectly. In most cases presented, BSD managed to purify the model successfully, leveraging the loss-guided splitting based on the Altruistic model.

It is worth noting that in experiments where the ground truth target class was 5 (dog), forcibly setting the pseudo-label to 3 (cat) led to a significant failure of the defense. This may be attributed to the inherent difficulty in distinguishing between these two classes. Furthermore, when faced with broader attack scenarios and dataset settings, relying solely on loss statistics may not be sufficient to ensure effective defense. Fortunately, our experiments demonstrate the strong robustness of the proposed Pseudo Target Approximation method. The OSS mechanism functioned as expected, enabling a resilient bi-perspective defense under challenging conditions.

**Insurance for the worst cases** We have intensively investigated our $y_t$ approximation method in both the main text (Section 5.4, Section 5.3, Section 5.5) and all the Appendix above. As the insurance, we have to state that, in the worst cases (if encountered), we could have the last resort that turns to a relatively weak assumption that is broadly applicable in common scenarios: In common scenarios where the dataset is a large but well-known benchmark dataset, the number of samples in each class is known to the public, and $y_t$ can be directly detected through label distribution.

### D.7. Resistance to Potential Adaptive Attacks

In the above experiments, we assume that attackers have no information about our backdoor defense. In this section, we consider a more challenging setting, where the attackers know the existence of our defense and can construct the poisoned dataset with an adaptive attack.

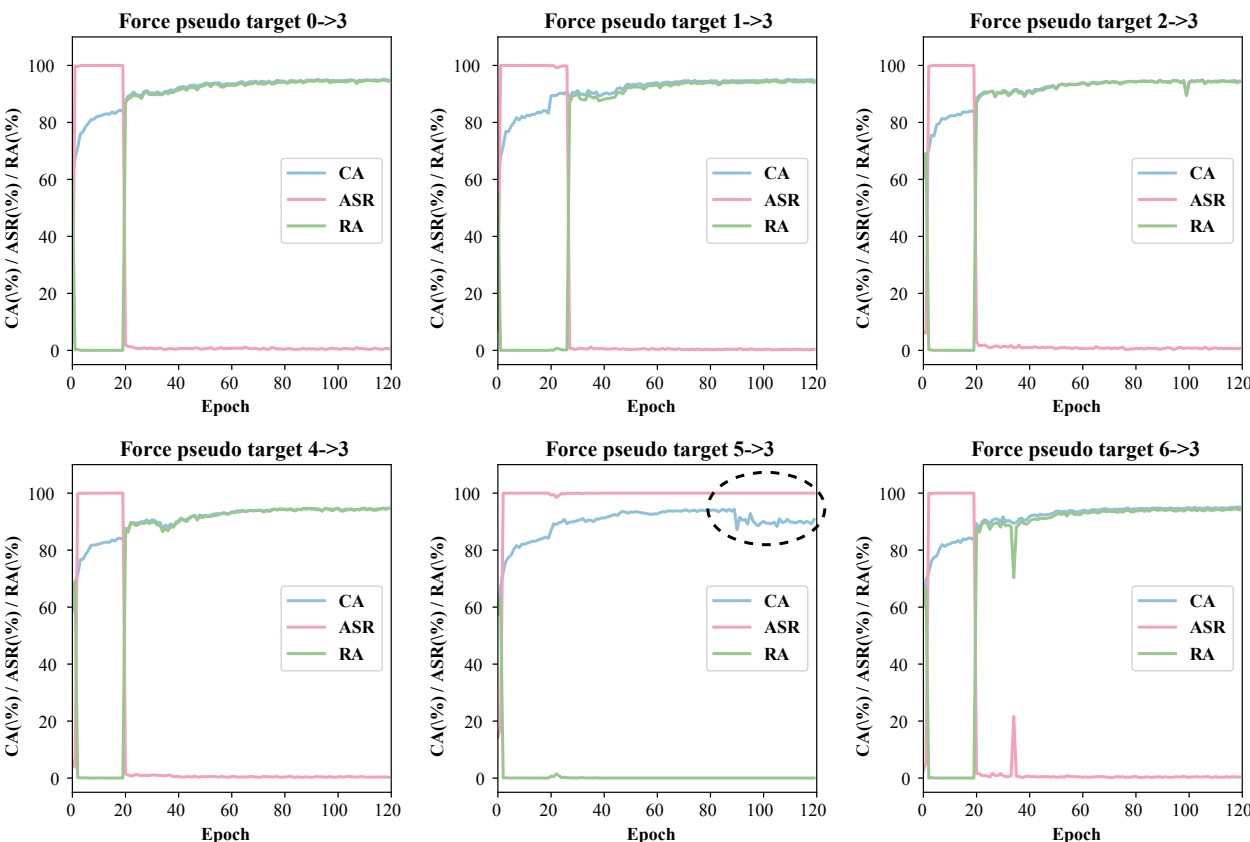

*Figure 10.* The clean accuracy (CA%), attack success rate (ASR%), and robust accuracy (RA%) of BSD when Forcing the pseudo target from x(ground truth) to 3.

**Threat model for the attackers.** Following existing work (Gao et al., 2023; Chen et al., 2017; Gu et al., 2017; Turner et al., 2018), we assume that the attackers can access the entire dataset and know the architecture of the victim model. However, the attackers can not control the training process after poisoned samples are injected into the training dataset.

**Method for adaptive attack.** Our method initializes the clean pool using bi-perspective splitting through OSS and ALS, which separate poisoned samples based on semantic and loss statistics, respectively. In general, there is a contradiction between increasing the loss values of poisoned samples (to bypass ALS) and achieving backdoor objectives. Furthermore, maintaining high semantic similarity to the target class (to bypass OSS) adds to the complexity. To craft such a trigger pattern that satisfies the above objectives, we use a PGD optimization to search for an average noise (among non-target classes) that is semantically close (judged by a proxy model) to the target class (to bypass OSS). Meanwhile, we control the $\ell_\infty$ ball bound as 8/255 and the poisoning rate as 0.01 to prevent it from being an obvious trigger that will be easily fitted (to bypass ALS).

**Settings.** We conduct experiments on CIFAR-10 with the following parameters: number of iterations, 15; step size, 1.5/255; perturbation magnitude, 8/255; trigger size, 32×32; and poisoning rate, 0.01. Although the attacker is assumed to have no knowledge of the model structure, we adopt a more challenging setting where the adversary uses the same model structure as the proxy model.

**Results.** The Clean Accuracy (CA) and Attack Success Rate (ASR) of this adaptive attack are 94.422% and 2.421%, respectively. While the ASR is slightly higher than that of other attacks on CIFAR-10, our defense clearly demonstrates strong resistance to the adaptive attack. Furthermore, when we increase the poisoning rate to 0.2 (20%), the CA and ASR remain at 91.040% and 0.903%, respectively, which is still within an acceptable range.

### D.8. Robustness against all2all attacks

All-to-all (all2all) attacks may pose challenges to certain components of our defense, particularly OSS and selective drop. However, all2all attacks are not typically considered essential scenarios in backdoor defense research currently (Li et al., 2021a; Huang et al., 2022; Zhu et al., 2023b; Guan et al., 2024; Zhang et al., 2023), for the following reasons: 1) The increased number of trigger-target pairs in all2all attacks requires significantly more training epochs for success. And all2all attacks reduce clean accuracy and exhibit slower convergence, making them easier to detect. ((Huang et al., 2024), Page 2: "As the number of classes increases, the accuracy and the attack success rate will decrease.") 2) Research on all2all attacks remains limited ((Li et al., 2022), Page 10: "However, there were only a few studies on all-to-all attacks. How to better design the all-to-all attack and the analysis of its properties remain blank."). 3) In practical applications, all2all attacks do not allow attackers to arbitrarily control predictions to specific targets, limiting their real-world threat.

Nevertheless, we still conducted supplementary experiments on BadNets with an all2all setting.

**Attack setting**: Following BadNets, with $y_t = (y + 1)\% n_c$, where $n_c$ is the number of classes.

**Defense setting**: To handle multiple target labels, BSD incurs additional computational costs by iterating through all classes as pseudo-targets during OSS. Clean indices from each pseudo-target are intersected to form the final OSS result. Additionally, we early stop at Stage 2 to avoid meaningless cost in Stage 3.

Since all-to-all attacks do not fundamentally change the nature of poison-label attacks, OSS remains effective for each individual classes. We visualized OSS spliting results in Figure 11, which reveals effective separation of clean samples of OSS. The CA, ASR, and DER performance are presented in Table 12, demonstrating a significant DER improvement compared to baseline methods. Notably, while BSD's ASR increases under all-to-all attacks, it effectively limits the attack success rate to the level of random prediction ($1/n_c = 10\%$).

In conclusion, our BSD method remains effective against all-to-all attacks. Furthermore, the OSS module can serve as a highly effective component for identifying clean samples in other backdoor defense methods.

*Table 12.* The clean accuracy (CA%), attack success rate (ASR%), and defense effective rating (DER%) of ASD and our BSD against BadNets-all2all on CIFAR-10.

| METHOD | BADNETS-ALL2ALL | | |
| --- | --- | --- | --- |
| | CA | ASR | DER |
| NO DEFENSE | 91.8 | 93.8 | - |
| ASD | 70.2 | 2.4 | 84.9 |
| BSD (OURS) | 91.2 | 10.5 | 91.3 |

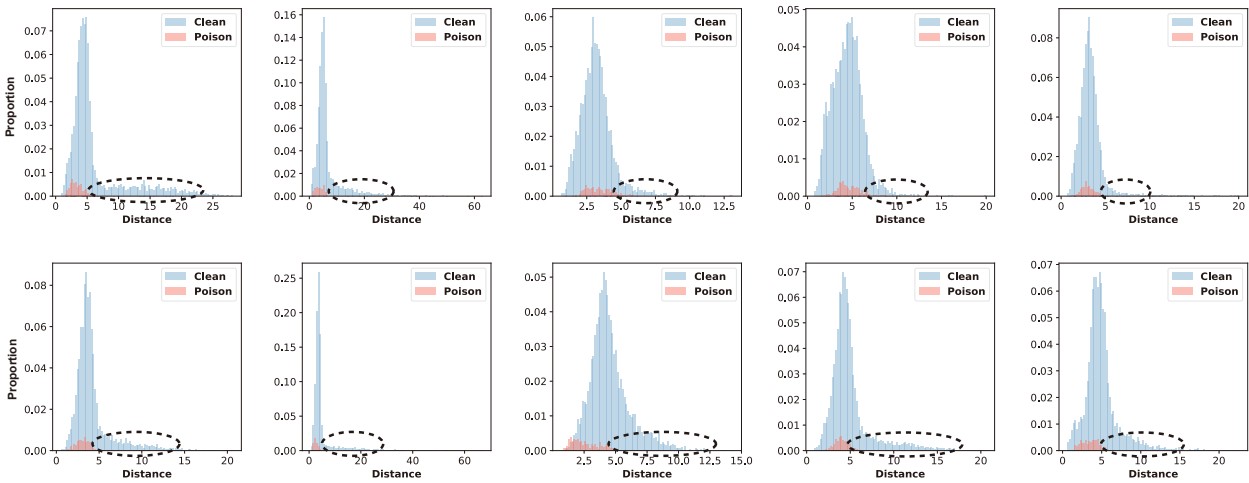

*Figure 11.* Visiualization fo the effectiveness of OSS against BadNet-all2all attack.

## D.9. Searching the best result of baselines

Notably, some baseline methods are sensitive to their hyper-parameter settings. The results reported in Table 1 represent their best performance obtained through grid search, as outlined in ASD (Gao et al., 2023). Similarly, for the additional attack settings, the results in Table 2 and Table 3 are based on their best outcomes (ranking on DERs) after grid search. For DBD, which is not sensitive to parameters, we report the best result using the default settings from BackdoorBench and the same configuration as in ASD.

*Table 13.* Grid search for FP against additional attacks on ResNet18 (Default represents the result under the default setting provided by backdoorbench).

| RATIO | LC | | SIG | | REFOOL | | NARCISSUS | |
|---|---|---|---|---|---|---|---|---|
| | CA | ASR | CA | ASR | CA | ASR | CA | ASR |
| DEFAULT | 87.1 | 24.4 | 87.1 | 60.8 | 86.5 | 23.0 | 87.2 | 63.4 |
| 0.1 | 87.3 | 79.8 | 87.2 | 81.4 | 86.8 | 25.6 | 87.4 | 72.4 |
| 0.2 | 87.0 | 59.4 | 87.0 | 82.4 | 86.5 | 28.0 | 86.7 | 77.8 |
| 0.3 | 86.7 | 51.6 | 87.0 | 83.2 | 86.5 | 27.6 | 87.2 | 73.9 |
| 0.4 | 87.0 | 49.0 | 87.2 | 85.7 | 86.4 | 28.8 | 87.2 | 77.6 |
| 0.5 | 85.7 | 67.3 | 86.2 | 86.1 | 85.2 | 29.6 | 86.7 | 79.0 |
| 0.6 | 86.0 | 73.6 | 86.7 | 90.1 | 85.6 | 31.6 | 86.6 | 80.7 |
| 0.7 | 86.3 | 80.0 | 86.8 | 88.5 | 86.1 | 31.0 | 87.1 | 79.6 |
| 0.8 | 87.0 | 74.9 | 86.9 | 87.7 | 86.4 | 28.3 | 87.3 | 78.5 |
| 0.9 | 87.3 | 62.0 | 87.0 | 80.3 | 86.5 | 25.2 | 87.6 | 72.3 |

*Table 14.* Grid search for FP against representative attacks on Mobilenetv2 (Default represents the result under the default setting provided by backdoorbench).

| RATIO | BADNET | | BLENDED | | WANET | |
|---|---|---|---|---|---|---|
| | CA | ASR | CA | ASR | CA | ASR |
| DEFAULT | 77.9 | 8.3 | 75.9 | 30.8 | 82.2 | 2.4 |
| 0.1 | 80.8 | 58.3 | 78.4 | 57.9 | 79.3 | 3.0 |
| 0.2 | 79.6 | 84.0 | 78.6 | 56.1 | 80.1 | 3.8 |
| 0.3 | 80.6 | 99.8 | 79.0 | 51.8 | 77.2 | 5.8 |
| 0.4 | 79.5 | 73.5 | 77.8 | 53.7 | 77.3 | 4.4 |
| 0.5 | 79.5 | 10.9 | 77.1 | 70.0 | 78.3 | 1.3 |
| 0.6 | 79.7 | 19.3 | 76.6 | 67.3 | 78.6 | 2.0 |
| 0.7 | 79.1 | 61.7 | 76.3 | 65.7 | 79.1 | 1.9 |
| 0.8 | 79.7 | 12.7 | 77.3 | 66.1 | 78.3 | 1.2 |
| 0.9 | 80.4 | 99.2 | 78.4 | 62.6 | 79.3 | 2.5 |

*Table 15.* Grid search for NAD against additional attacks on ResNet18 (Default represents the result under the default setting provided by backdoorbench).

| BETA | LC | | SIG | | REFOOL | | NARCISSUS | |
|---|---|---|---|---|---|---|---|---|
| | CA | ASR | CA | ASR | CA | ASR | CA | ASR |
| DEFAULT | 86.2 | 69.7 | 86.0 | 83.6 | 85.3 | 49.3 | 86.1 | 84.9 |
| 100 | 87.3 | 98.7 | 87.1 | 95.9 | 86.5 | 66.9 | 86.8 | 93.3 |
| 500 | 86.2 | 69.7 | 86.0 | 83.6 | 85.3 | 49.3 | 86.1 | 84.9 |
| 1000 | 86.2 | 69.7 | 86.0 | 83.6 | 85.3 | 49.3 | 86.1 | 84.9 |
| 1500 | 86.2 | 69.7 | 86.0 | 83.6 | 85.3 | 49.3 | 86.1 | 84.9 |
| 2000 | 86.2 | 69.7 | 86.0 | 83.6 | 85.3 | 49.3 | 86.1 | 84.9 |
| 2500 | 86.2 | 69.7 | 86.0 | 83.6 | 85.3 | 49.3 | 86.1 | 84.9 |
| 5000 | 86.2 | 69.7 | 86.0 | 83.6 | 85.3 | 49.3 | 86.1 | 84.9 |
| 7500 | 86.2 | 69.7 | 86.0 | 83.6 | 85.3 | 49.3 | 86.1 | 84.9 |
| 1.E+04 | 86.2 | 69.7 | 86.0 | 83.6 | 85.3 | 49.3 | 86.1 | 84.9 |
| 1.E+05 | 86.2 | 69.7 | 86.0 | 83.6 | 85.3 | 49.3 | 86.1 | 84.9 |
| 1.E+06 | 86.2 | 69.9 | 85.8 | 83.3 | 85.3 | 49.3 | 86.2 | 84.9 |
| 1.E+07 | 85.9 | 50.5 | 85.7 | 84.5 | 85.5 | 46.5 | 86.5 | 81.0 |
| 1.E+08 | 86.0 | 73.9 | 85.8 | 88.7 | 85.6 | 42.9 | 86.1 | 83.2 |
| 1.E+09 | 85.3 | 68.2 | 85.7 | 87.0 | 85.6 | 44.8 | 86.5 | 84.1 |
| 1.E+10 | 85.9 | 69.5 | 85.8 | 83.0 | 85.4 | 47.7 | 86.0 | 86.2 |
| 1.E+11 | 85.9 | 65.6 | 85.7 | 86.3 | 85.6 | 42.5 | 86.4 | 83.4 |

*Table 16.* Grid search for NAD against representative attacks on Mobilenetv2 (Default represents the result under the default setting provided by backdoorbench).

| BETA | BADNET | | BLENDED | | WANET | |
|---|---|---|---|---|---|---|
| | CA | ASR | CA | ASR | CA | ASR |
| DEFAULT | 78.5 | 11.7 | 76.2 | 51.6 | 81.1 | 4.2 |
| 100 | 79.7 | 99.1 | 79.5 | 56.8 | 77.3 | 3.4 |
| 500 | 78.5 | 11.7 | 76.2 | 51.6 | 81.1 | 4.2 |
| 1000 | 78.5 | 11.7 | 76.2 | 51.6 | 81.1 | 4.2 |
| 1500 | 78.5 | 11.7 | 76.2 | 51.6 | 81.1 | 4.2 |
| 2000 | 78.5 | 11.7 | 76.2 | 51.6 | 81.1 | 4.1 |
| 2500 | 78.5 | 11.7 | 76.2 | 51.6 | 81.1 | 4.1 |
| 5000 | 78.5 | 11.7 | 76.2 | 51.6 | 81.1 | 4.1 |
| 7500 | 78.5 | 11.7 | 76.2 | 51.6 | 81.1 | 4.1 |
| 1.E+04 | 78.5 | 11.7 | 76.2 | 51.6 | 81.1 | 4.1 |
| 1.E+05 | 78.6 | 29.1 | 76.1 | 51.3 | 79.8 | 4.0 |
| 1.E+06 | 78.5 | 23.0 | 75.9 | 49.4 | 81.5 | 3.2 |
| 1.E+07 | 79.3 | 25.4 | 76.8 | 47.6 | 81.0 | 3.8 |
| 1.E+08 | 78.8 | 14.8 | 76.8 | 58.9 | 81.0 | 3.2 |
| 1.E+09 | 78.9 | 17.9 | 76.0 | 46.0 | 80.7 | 5.4 |
| 1.E+10 | 79.2 | 25.0 | 76.5 | 51.8 | 81.6 | 4.1 |
| 1.E+11 | 78.8 | 41.5 | 76.7 | 56.9 | 81.3 | 4.3 |

*Table 17.* Grid search for ABL against additional attacks on ResNet18 (Default represents the result under the default setting provided by backdoorbench).

| GAMMA | LC | | SIG | | REFOOL | | NARCISSUS | |
|---|---|---|---|---|---|---|---|---|
| | CA | ASR | CA | ASR | CA | ASR | CA | ASR |
| DEFAULT | 71.1 | 4.3 | 37.5 | 0.0 | 63.3 | 94.0 | 62.7 | 0.4 |
| 0.0 | 81.2 | 7.3 | 81.2 | 74.6 | 79.4 | 86.4 | 80.7 | 76.7 |
| 0.1 | 81.2 | 6.0 | 79.7 | 41.5 | 72.0 | 93.5 | 80.3 | 35.1 |
| 0.2 | 80.3 | 3.8 | 73.9 | 16.5 | 76.3 | 82.0 | 79.4 | 32.7 |
| 0.3 | 78.1 | 0.9 | 76.1 | 14.4 | 76.0 | 86.5 | 79.3 | 7.1 |
| 0.4 | 80.2 | 1.6 | 67.6 | 5.1 | 71.4 | 85.4 | 76.6 | 44.1 |
| 0.5 | 78.7 | 1.0 | 67.0 | 6.9 | 75.2 | 93.8 | 78.3 | 17.5 |

*Table 18.* Grid search for ABL against representative attacks on Mobilenetv2 (Default represents the result under the default setting provided by backdoorbench).

| GAMMA | BADNET | | BLENDED | | WANET | |
|---|---|---|---|---|---|---|
| | CA | ASR | CA | ASR | CA | ASR |
| DEFAULT | 68.0 | 24.4 | 67.3 | 2.6 | 50.9 | 0.5 |
| 0.0 | 81.3 | 36.9 | 77.3 | 21.6 | 68.9 | 74.8 |
| 0.1 | 81.3 | 36.9 | 77.3 | 21.6 | 68.9 | 74.8 |
| 0.2 | 81.3 | 36.9 | 77.3 | 21.6 | 68.9 | 74.8 |
| 0.3 | 81.3 | 36.9 | 77.3 | 21.6 | 68.9 | 74.8 |
| 0.4 | 79.7 | 13.6 | 78.4 | 15.1 | 68.9 | 74.8 |
| 0.5 | 81.3 | 48.1 | 80.4 | 33.4 | 68.9 | 74.8 |

*Table 19.* Result of DBD against additional attacks on ResNet18 (Default represents the result under the default setting provided by backdoorbench, Default2 represents the recommended setting used in ASD).

| SETTING | LC | | SIG | | REFOOL | | NARCISSUS | |
|---|---|---|---|---|---|---|---|---|
| | CA | ASR | CA | ASR | CA | ASR | CA | ASR |
| DEFAULT | 83.2 | 98.1 | 77.6 | 99.9 | 87.0 | 0.1 | 87.3 | 99.6 |
| DEFAULT2 | 82.46 | 99.42 | 80.12 | 99.9 | 90.84 | 2.34 | 80.73 | 99.61 |

*Table 20.* Result of DBD against representative attacks on Mobilenetv2 (Default represents the result under the default setting provided by backdoorbench, Default2 represents the recommended setting used in ASD).

| SETTING | BADNET | | BLENDED | | WANET | |
|---|---|---|---|---|---|---|
| | CA | ASR | CA | ASR | CA | ASR |
| DEFAULT | 65.5 | 0.0 | 69.0 | 0.0 | 58.4 | 12.4 |
| DEFAULT2 | 54.34 | 0 | 64.22 | 0 | 57.22 | 14.11 |

