# OpenReview forum: "Bi-perspective Splitting Defense: Achieving Clean-Seed-Free Backdoor Security"
_ICML.cc/2025/Conference — ICML 2025 poster_

### Official Review · Reviewer_FTAm · 2025-03-12

**Overall Recommendation:** 2

**Summary:**

This paper addresses poisoning backdoor attacks. Specifically, the authors aim to challenge the assumption of accessing clean data in current backdoor defense literature, with the main idea of utilizing both easier-to-obtain target labels and clean, hard samples. They propose a Bi-perspective Splitting Defense (BSD) which relies on semantic and loss statistics through OSS and ALS, respectively. The proposed method is evaluated on benchmark datasets to demonstrate its effectiveness.

## Update after rebuttal.

I thank the authors for their rebuttals. It appears that my original review was accurate in stating:

> implicitly assumes the existence of a feature that can reliably distinguish clean data from backdoor data.

As a result, I find it quite confusing to understand how your approach can be described as clean-data-free. I strongly recommend revising the wording to reflect this more accurately.

Regarding the experiments, I do not find them comprehensive enough—especially concerning the attack part—given the vast body of literature on backdoor attacks, as I mentioned in [2].

With that being said, I will adjust my rating to a borderline score.

**Claims And Evidence:**

The claims are supported by either textual and mathematical elaborations or empirical evidence.

**Essential References Not Discussed:**

I have listed some missing references in previous sections.

**Experimental Designs Or Analyses:**

Given the extensive body of literature on backdoor defenses, the evaluations in the current version require significant improvement. For instance, the number of evaluated attacks and defenses is not comprehensive enough. I suggest that the authors follow the setup in [2], where more than 10 attacks and 7 defenses were tested.

Refs: [2] https://arxiv.org/abs/2205.13616

**Methods And Evaluation Criteria:**

I believe the paper has some methodological weaknesses, which I will elaborate on below.

- **Free clean-data assumption**: First, I find this assumption invalid. Theoretically, it can be shown (and I can provide a proof if the authors are interested) that **without any information** about the clean data, it is impossible to perform detection or filtering-based defenses. In this sense, the assumption effectively states that, although direct access to clean data is unavailable, a proxy for it is accessible.

In particular, while your method operates on mixed data, it **implicitly assumes the existence of a feature that can reliably distinguish clean data from backdoor data**. This idea has already been extensively explored in previous literature [1]. In other words, the paper does not address the scenario where no clean data is available at all.

Building upon the previously mentioned implicit assumption, the proposed bi-perspective approach does not appear novel to me. Overall, I find the paper's novelty and originality to be limited.

Refs: [1] https://arxiv.org/abs/1811.00636

**Other Comments Or Suggestions:**

- Please check my previous comments

**Other Strengths And Weaknesses:**

- Please check my previous comments

**Questions For Authors:**

- Please check my previous comments

**Relation To Broader Scientific Literature:**

Backdoor attacks and defenses fall within the broader domain of core machine learning.

**Theoretical Claims:**

There is no theoretical proof provided. However, the paper includes some mathematical derivations, which I believe are correct.

---

> ### Author Rebuttal · Authors · 2025-04-01
>
> Thank you for your comment.
>
> # R1 Assumption&Novelty
>
> The so-called implicit assumption that `implicitly assumes ... backdoor data` was not a weakness of our paper, and it does not influence the novelty when taking [1] into account. See below for details.
>
> ## R1.1 Clarification
>
> As the reviewer may have interpreted the background differently, we would like to re-clarify the defense scenario. Our focus is on a practical scenario where defenders attempt to train a benign model using partially poisoned backdoor datasets without **known to be clean** subsets (**See lines 41-76, Introduction; lines 94-121, Preliminary**).
>
> To formally define, the defender has a training set $D = D_{c} \cup D_p$ , but the defender doesn't know which part of the samples is clean, and has neither known clean samples $D_{c*}$($D_{c*} \subset D_c$) nor external clean samples $D_{extra}$($D_{extra}\cap D=\varnothing$).
>
> In our previous submission, we consistently use the terms *clean subsets / extra clean subsets / clean seed* rather than *clean data* throughout the main text and explicitly formulate the training set as $D = D_{c} \cup D_p$ instead of $D = D_p$ (**Line 107, Preliminary**). While other reviewers have not raised concerns regarding this aspect, we appreciate the opportunity to refine our explanation and will further clarify this definition in our revision.
>
> ## R1.2 Free clean-data assumption
>
> As we re-clarified above, we didn't make such an assumption as you interpreted, thus it does not constitute a weakness.
>
> To understand why our BSD works on mixed datasets without known-clean samples, that's because we leverage general attack-related priors, including: **1)** From the perspective of loss statistics, neural networks tend to overfit backdoor samples, resulting in lower loss values(ALS); **2)** From the perspective of semantic information, identifying poisoned samples from backdoor attacks can be reframed as an open-set recognition problem(OSS). (**See lines 152-160, page 3**)
>
> For your claim that `without any information ... filtering-based defenses`, there may be a misunderstanding regarding our background:
>
> - It seems that you are referring to a scenario where there is absolutely no clean data in the training set, **it's indeed an impossible task**. But this scenario is not what our paper focuses on, “no clean data at all” is quite different from the “no additional clean subsets” scenario we address. Your comment below, `In other words ... no clean data is available at all` seems to suggest that we should solve a task that just described as impossible.
>
> - > If you mean that the defender has no assumptions regarding clean data information but is dealing with a mixed dataset, whether defense is possible depends on how you define “clean data information” and whether general attack-related priors count. Your [1] is a good case. It proposes a method that does not require known clean subsets but distinguishes between samples within a mixed dataset. It managed this using attack-related priors, treating poisoned samples as the minorities within each category.
>
> ## R1.3 Novelty
>
> Our novelty is already claimed (**lines 75-98, Introduction**), and Ref[1] doesn't affect our novelty, in detail:
>
> - While [1] also assumes no extra clean sets, the core defense mechanism differs. It detects poisoned samples as intra-class outliers, considering **only** the samples within each current class. In contrast, our OSS module innovatively reframes the problem based on open-set recognition, OSS **jointly** considers the target class and all other classes for effective distinction. After identifying $y_t$ and warming up the main model, the OSS module distinguishes samples based on different feature distances between the target class $D_{t}$ (i.e., UKCs + UUCs) and the remaining classes $D_{nt}$ (i.e., KKCs). (Details in **Section 4.1.1**)
> - [1] relies on the assumption that **poisoned samples are intra-class minorities**. As the intra-class proportion of poison samples($\approx\rho$) increases, class-wise mean representation goes closer to poison samples, reducing detection effectiveness. When poisoned samples became the intra-class majority (for large $\rho$ values or in imbalanced datasets like GTSRB), clean samples became outliers instead. In contrast, our method is robust against this issue and is verified by experiments under a larger $\rho$ range (**Figure 3, page 8**).
> - Additional differences in other aspects: adaptive vs. one-time partitioning and a semi-supervised framework vs. detect-and-retrain framework, etc.
>
> ---
>
> # R2 Additional Experiments
>
> Our experiments already cover 7 attacks in the main text and more attack variants in the appendix, along with 6 defenses and additional recent defenses like VaB, D-ST, and D-BR. **This is sufficiently comprehensive since** 1) the threat model specifies poisoning-based attacks; 2) related works like ABL and DBD evaluate fewer cases. **More results in https://postimg.cc/DSn6S9nG.**

---

### Official Review · Reviewer_apHg · 2025-03-14

**Overall Recommendation:** 2

**Summary:**

This paper proposes a backdoor attack defense method, Bi-perspective Splitting Defense (BSD), which does not rely on additional clean subsets. BSD utilizes semantic characteristics through open set recognition-based splitting (OOS) and loss statistics characteristics through altruistic model-based data splitting (ALS) to distinguish clean samples. In their experiments, BSD demonstrates its superiority compared to five state-of-the-art defenses.

## update after rebuttal

Thanks for the authors' feedback. After reading the response, I would like to maintain my original score.

**Claims And Evidence:**

The authors define backdoor samples as Unknown Classes (UKCs) in their proposed BSD. However, they also admit that clean-label attacks are not UKCs, which makes me question the soundness of their defense claims. This raises doubts about whether their definition is accurate enough—doesn't this create an inconsistency?

If clean-label attacks are not UKCs, it implies that OSS is useless against clean-label attacks. The authors also claim that ALS is strong enough to defend against clean-label attacks, so why not simply use ALS for splitting? If that's the case, would applying ALS alone be effective against all types of backdoor attacks? And if so, is OSS even necessary?

**Essential References Not Discussed:**

The key contribution is the proposal of a high-performance backdoor defense that does not rely on a clean subset. However, the paper only cites some backdoor defenses with worse performance while overlooking a state-of-the-art approach that also does not require a clean subset—namely, Progressive Isolation of Poisoned Data (PIPD), which was published in AAAI 2024.

**Experimental Designs Or Analyses:**

Yes, I believe this work requires more experiments to demonstrate its soundness and validity. In the case of clean-label attacks, semi-supervised learning lacks the ability to correct mislabeled samples. Both DBD and ASD incorporate semi-supervised learning in their defense mechanisms, yet they struggle to defend against clean-label attacks. So why would BSD be effective against them?
For clean-label attacks, it is evident that OSS is ineffective because, in such attacks, poisoned samples are also UUCs (Unknown Unknown Classes). However, the authors claim that ALS alone is strong enough to separate poisoned samples in clean-label attacks. This assertion requires an ablation study to confirm whether ALS alone is truly capable of achieving this. Moreover, if ALS can effectively defend against clean-label attacks, its effectiveness against poisoned-label attacks should also be evaluated.

Additionally, the authors should analyze the performance of the target label select mechanism, as it raises concerns about the reliability of target label distinction. Given that clean-label attacks are notoriously difficult to defend against, why did BSD only get evaluated on a single benchmark dataset?

Furthermore, the study lacks experiments evaluating the defense performance across different model architectures, which is crucial for demonstrating the robustness and generalizability of BSD.

**Methods And Evaluation Criteria:**

Based on the results of their experiments, they did propose a promising backdoor defense. However, the experimental validation is not robust enough to convincingly demonstrate its effectiveness.

**Other Comments Or Suggestions:**

None.

**Other Strengths And Weaknesses:**

Other Strengths: Framing the backdoor attack problem as an open-set recognition issue offers a novel perspective.
Other Weaknesses: The writing is confusing; the concepts of Known Unknown Classes (KUCs) and Known Known Classes (KKCs) are not clearly introduced, and the figures are also quite disorganized.

**Questions For Authors:**

Q1. How effective is the experimental method for determining y_target? Can it reliably identify y_target? How does this method perform in attack scenarios where the number of poisoned samples is relatively small?

Q2. Clearly, in the case of a clean-label attack, OOS becomes ineffective since the poisoned samples in clean-label attacks do not have modified labels. However, the authors claim that the ALS stage alone can achieve effective separation. If that is the case, why was there no ablation study on BSD to verify whether the ALS stage truly performs as well as claimed? If ALS alone is sufficient, should it also be capable of defending against poison-label attacks? Is it really as effective as the authors suggest?

**Relation To Broader Scientific Literature:**

The key contributions of the paper are closely related to the security of artificial intelligence.

**Theoretical Claims:**

Yes, for clean-label attacks, the theoretical foundation of OSS does not hold!

---

> ### Author Rebuttal · Authors · 2025-04-01
>
> Thank you for your comments.
>
> **Extended Figure/Tables in anonymous link: https://postimg.cc/68rDsgN3.**
>
> # R1 Understanding BSD's Robustness Against Clean-Label Attacks
>
> First, in response to `Given that clean-label...single benchmark dataset?`, we provided the experiment results of BSD on GTSRB (**Extended Table 1**).
>
> The primary reason BSD resists clean-label backdoor attacks: **1) MixMatch could properly process unlabeled poisoned clean-label data; 2) BSD effectively split poisoned samples.** Specifically, MixMatch's mixup operation visually weakens the trigger in the unlabeled data, preventing trigger-target label association. So, as long as the poisoned samples are correctly placed into the poison pool (unlabeled data), BSD effectively mitigates their impact.
>
> **We start by investigating semi-supervised involved defenses like ASD and DBD.** These works report counterintuitive good performance against clean-label attacks. DBD included a brief explanation of its effectiveness against clean-label attacks (in their Appendix O, page25). Reproducing ASD showed that failures occurred when poison samples remained in the clean pool. Reviewing logs from two failed ASD experiments, we found many poison samples misclassified into the clean pool(**Extended Table 2**).
>
> **This led us to hypothesize that MixMatch could properly process unlabeled poisoned clean-label data, but DBD and ASD failed for not robust split methods instead.**
>
> Further investigation highlighted MixMatch’s mixup operation as critical. MixMatch removes original labels and mixes multiple inputs on the image level, effectively weakening the visual impact of the trigger:
>
> $$ \tilde{x} = \lambda' x_l + (1 - \lambda') x_u, \quad \tilde{y} = \lambda' y_l + (1 - \lambda') y_u, \quad\lambda\sim Beta(\alpha,\alpha),\quad \lambda'=\text{max}(\lambda,1-\lambda). $$
>
> With $\alpha=0.75$ by default, expectation of $\lambda'$ ≈0.78, reducing trigger prominence in $x_u$. We visualize the mixed samples in **Extended Figure 1** to better present this operation. Plus, we follow ASD set a 5x smaller $\lambda_u$, further reducing the influence of unlabeled data and mitigating clean-label attacks.
>
> **To experimentally verify this hypothesis, we enforced a secure clean-poison split where no poison samples were included in the clean pool.** Under this condition, MixMatch effectively nullified the impact of clean-label attacks, as shown in **Extended Table 3**.
>
> **That said, it's still a necessity that we do not split poison samples into clean pools.** Therefore, since the final model does not collapse (Table 2, page6), it supports the reliability of ALS’s split results. Moreover, while OSS seems ineffective against clean-label poisoned samples, it still serves two important purposes: 1) It provides a secured warm-up phase for the main model, preventing an immediate collapse (see our definition in Appendix C.3); 2) Although OSS may not directly identify poison samples, it selects samples from $D_t$ that are farthest in feature space from those in $D_{nt}$, i.e., representitive samples of $D_t$. This result intersects with the clean pool identified by ALS, improving the quality of init clean samples. Therefore, our statement is that ALS “compensates for the limitations of OSS”(line 375, page7), instead of ALS alone is sufficient.
>
> ---
>
> # R2 Important Baseline
> Respecting the reviewers’ suggestions, we made efforts to implement PIPD(**Extended Table 4**).
>
> Note that we did not intentionally avoid citing higher-performing works. The exclusion of PIPD (AAAI24) was due to:
>
> - We conduct reproductive evaluations before adding any baselines, and PIPD's reproductivity is poor. Specifically, across its paper, appendix, and GitHub repository, we found many essential code/parameters/setting omissions.
> - Methodologically, while PIPD’s approach is intuitively good, it relies on good starts from pre-isolation using LGA. However, our analysis suggests LGA alone may require attack-specific adjustments to hyperparameters (e.g., optimizer choice/learning rates). That's also why we replaced our early version of LGA-based $y_t$ estimation.
>
> ---
>
> # R3 Estimation of y_t
>
> We would like to emphasize that the estimation method for $y_t$ is not our major contribution, and we have verified its robustness(Table 4, page7; Figure 9, page19; Table 8, page19; Table 11, page20) and discussed multiple insurance measures(line 1011-1032, page19). To answer how this method performs when the number of poisoned samples is relatively small, see Figure 3 (a), page8.
>
> ---
>
> # R4 Others
>
> Model structures: Since we have no model-specific assumptions, our method is inherently model-agnostic. Additionally, our experiments on MobileNet further support this claim(Section 5.3). We also add a brief experiment in **Extended Table 5**.
>
> Presentation: We appreciate your insights about strengthening the presentation of our work. We will update our modifications accordingly in our revision (once we get a chance).

---

### Official Review · Reviewer_Mmmu · 2025-03-14

**Overall Recommendation:** 4

**Summary:**

This paper introduces ​Bi-perspective Splitting Defense (BSD), a novel in-training backdoor defense framework designed to train robust models from poisoned datasets without requiring clean data. By integrating semantic and loss-based perspectives, BSD addresses critical limitations in existing defenses, particularly their reliance on impractical clean subsets or computationally expensive feature analysis, by proposing the OSS and ALS techniques for splitting datasets. The experimental results validate its effectiveness across various settings.

**Claims And Evidence:**

The claims in this paper are supported by convincing evidence.

- Open-set recognition task and poison sample detection are similar.
    - Section 4.1.1 discusses the open set recognition setting and the relationship to poison and clean sample detection in detail.

**Essential References Not Discussed:**

It seems that the paper [1] on in-training defense with similar data isolation techniques is not compared and discussed as the baseline. The latest defense method *ASD* is from CVPR-23, which is behind the current SOTA method.

[1] Progressive Poisoned Data Isolation for Training-Time Backdoor Defense, AAAI-24.

**Experimental Designs Or Analyses:**

The experiments are comprehensive and the experimental settings are thoroughly illustrated in this paper.

**Methods And Evaluation Criteria:**

This paper has no problem in the method and evaluation following the previous settings.

**Methods**:
- The proposed methods make sense for the problem, where the feature distance, target-label approximation, and loss-based sample selection techniques are commonly used in the literature for backdoor learning.
- The in-training defense targeting on splitting the dataset for training is reasonable and practical.

**Evaluation**:
- The evaluation metrics (CA, ASR, and DER) and three benchmark datasets are considered adequate in the literature to fully evaluate clean and backdoor performance.

**Other Comments Or Suggestions:**

None.

**Other Strengths And Weaknesses:**

Pros:

- The paper is well-structured and easy-to-understand with dedicated figures.
- The extensive experiments on the method details are conducted thoroughly.
- The defense performances are promising in both CA and ASR.

Cons:
- See the *Essential References Not Discussed* above.

**Questions For Authors:**

None.

**Relation To Broader Scientific Literature:**

The relation to the literature is well elaborated in this paper.

**Theoretical Claims:**

There is no theoretical claim in this paper.

---

> ### Author Rebuttal · Authors · 2025-04-01
>
> **Thank you for your constructive feedback and for recommending our paper for acceptance. We sincerely appreciate your thorough review and valuable insights. In response to your remaining concern, we present more baselines.**
>
> # Additional results
> As you suggested, which aligns with other reviewers, we added the PIPD as an additional baseline, together with NAB [1] and NPD [2]. As presented in **Extended Table 1**, while these methods demonstrated exceptional defense performance against specific attacks, our implementation suggests that they fail to achieve consistent defense across all seven attacks under a single set of parameter settings. We believe that with attack-specific parameter tuning, they could potentially achieve the performance claimed in their paper. However, we must emphasize that such attack-specific tuning is impractical for defenders in real-world scenarios.
>
> **Extended Table 1. Addition baselines on CIFAR-10. Since the table is too large, we present the full version in this anonymous link: https://postimg.cc/ftmSjbvR.** **PIPD1** uses the Adam optimizer (learning rate = 1e-3), **PIPD2** resets the optimizer (clearing momentum and resetting the learning rate) before the Selective Training stage, and **PIPD3** applies both a larger penalty ($\lambda = 5$) and the optimizer reset.
> | Attack |  <   | BadNet |  >   |  <   | Blend |  >   |  <   | WaNet |  >   |  <   | Refool |  >   |  <   |  LC  |  >   |  <   | SIG  |  >   |  <   | Narcissus |  >   | Avg  | Avg  |
> | :----: | :--: | :----: | :--: | :--: | :-----: | :--: | :--: | :---: | :--: | :--: | :----: | :--: | :--: | :--: | :--: | :--: | :--: | :--: | :--: | :-------: | :--: | :--: | :--: |
> | Metric |  CA  |  ASR   | DER  |  CA  |   ASR   | DER  |  CA  |  ASR  | DER  |  CA  |  ASR   | DER  |  CA  | ASR  | DER  |  CA  | ASR  | DER  |  CA  |    ASR    | DER  | ASR  | DER  |
> | PIPD1  | 86.8 |  24.8  | 83.6 | 86.8 |  88.5   | 51.3 | 95.9 |  0.2  | 99.9 | 79.4 |  61.9  | 61.5 | 87.5 | 89.7 | 49.0 | 81.0 | 88.7 | 48.0 | 84.2 |   36.2    | 76.1 | 55.7 | 67.1 |
> | PIPD2  | 78.9 |  36.0  | 74.0 | 83.5 |  55.7   | 66.0 | 94.3 |  0.5  | 99.7 | 76.6 |  70.6  | 55.7 | 73.3 | 29.6 | 72.0 | 80.3 | 87.6 | 48.2 | 73.3 |   29.6    | 73.9 | 44.2 | 69.9 |
> | PIPD3  | 85.0 |  13.1  | 88.5 | 73.6 |   6.4   | 85.7 | 94.1 |  0.2  | 99.8 | 67.5 |  62.2  | 55.4 | 70.9 | 12.3 | 79.4 | 72.8 | 85.5 | 45.6 | 75.8 |   35.0    | 72.5 | 30.7 | 75.3 |
> |  BSD   | 95.1 |  0.9   | 99.6 | 94.9 |   0.8   | 98.8 | 94.5 |  0.8  | 99.6 | 92.4 |  1.2   | 98.4 | 93.8 | 0.0  | 97.0 | 94.8 | 0.5  | 99.0 | 94.3 |    0.0    | 99.3 | 0.6  | 98.8 |
> # Implementation Description
> The rationale for excluding PIPD in the previous submission:
> - Methodologically, while PIPD's approach appears intuitively feasible, it relies on good starts from Pre-isolation using Local Gradient Ascent (LGA), proposed in ABL. Facing varying backdoor attacks, we found that it may require attack-specific tuning of hyper-parameters, which is not allowed for defenders. This concern, also supported by our early experimental observations, was one of the reasons we ultimately replaced our old version $y_t$ estimation (basically based on LGA).
> - Before incorporating every method as a baseline, we conduct preliminary experimental validation to assess its reproductivity, such as evaluating its defense performance on the typical CIFAR10-BadNet combination. However, we observed that PIPD has not provided comprehensive open-source code. Specifically, after reviewing the main text, appendix, and GitHub repository of PIPD, we found PIPD: **(1)** provides empty README files and has no demos; **(2)** does not include any attack implementation (not even BadNet); **(3)** does not provide the implementation of Pre-isolation and Selective training stage; and **(4)** omits several critical training parameters, including optimizer selection, learning rate, the choice of the $\gamma$ in the Pre-isolation stage, the choice of $\lambda$ in selective training, and whether the optimizer/learning rate needs any resetting during the training.
>
> Settings:
> - To ensure experimental fairness, we merge the poisoned training data from BackdoorBench to generate datasets aligning with PIPD’s PyTorch dataset definition.
> - For the non-open-sourced components of PIPD, we implement them based on their pseudocode and equations. For key parameters not explicitly specified in PIPD, we set $\gamma = 0.5$ in LGA for Pre-isolation, as recommended in ABL, and applied a default penalty of $\lambda = 1$ in Selective Training.
> - Besides PIPD, we add two more baselines, namely NAB and NPD, with NAB being an in-training backdoor defense and NPD being a recent post-training defense.
> ---
> \[1\]: Liu M, Sangiovanni-Vincentelli A, Yue X. Beating backdoor attack at its own game, ICCV 2023.
>
> \[2\]: Zhu M, Wei S, Zha H, et al. Neural polarizer: A lightweight and effective backdoor defense via purifying poisoned features, NIPS 2023.

---

### Official Review · Reviewer_2zkR · 2025-03-14

**Overall Recommendation:** 4

**Summary:**

This paper introduces a clean-data-free method, named BSD, to defend against backdoor attacks in deep neural networks. BSD employs two complementary perspectives: Open Set Recognition-based Splitting (OSS) which uses semantic information, and Altruistic Model-based Loss Splitting (ALS) which leverages loss statistics. The approach includes pool initialization, class completion, and selective dropping strategies. Extensive experiments across multiple datasets and attack types show BSD achieves an average 16.29% improvement in Defense Effectiveness Rating compared to five state-of-the-art methods, while demonstrating robustness to different model architectures, poisoning rates, and target labels.

## Update after rebuttal.
The explanation of the author has resolved my concern and I have also read the commet of other reviewer. The novelty may indeed be a weakness.  But I will keep my score of Accept since I did not found any major defects.

**Claims And Evidence:**

The paper's core claims about BSD's effectiveness are generally well-supported by extensive experimental evidence across multiple datasets, attack types, and comparison with state-of-the-art methods. However, some claims lack sufficient support:

- The paper asserts that the bi-perspective approach is superior but doesn't fully explore why these two specific perspectives are optimal compared to other possible perspective combinations—such as those involving gradient-based metrics, alternative feature distributions, or multi-modal representations.

- BSD emphasizes that it does not rely on additional clean data and instead achieves precise segmentation of clean samples through OSS and ALS. Although the paper circumvents the need for external clean data by initializing a pseudo-clean pool based on OSS and ALS, the estimation process for the target label (yt) and the introduction of the altruistic model involve implicit assumptions; if, in certain scenarios, the target label is misestimated or the sample distribution deviates significantly from that of standard datasets, the segmentation performance may deteriorate, ultimately affecting the overall defense efficacy.

Despite these limitations, the extensive experimental results across various settings do provide substantial evidence for BSD's effectiveness as a clean-data-free backdoor defense method.

**Essential References Not Discussed:**

The paper adequately covers the essential references needed to understand the context for its key contributions.

**Experimental Designs Or Analyses:**

In this paper, the experimental design and analysis validate the effectiveness and robustness of the BSD method from multiple perspectives. This is accomplished by evaluating the approach on several datasets (such as CIFAR-10, GTSRB, and ImageNet) and using a variety of mainstream network architectures (including ResNet-18 and MobileNet-v2), as well as by testing against an array of backdoor attacks—namely BadNets, Blended, WaNet, Label-Consistent, SIG, Refool, Narcissus, and even clean-label and all-to-all attacks. Furthermore, experiments conducted under different poisoning rates and target label settings demonstrate that the method can consistently maintain high clean accuracy (CA) and a low attack success rate (ASR). Additionally, ablation studies and hyperparameter grid searches on key components—such as OSS (semantic-based sample splitting), ALS (loss-statistics-based sample splitting), class completion, and selective dropping—confirm the critical contributions of these modules to improving the defense effectiveness (DER), while training cost evaluations underscore the practical advantages of the method. **However, the results also indicate that setting certain hyperparameters to extreme values can significantly degrade performance, and the method’s reliance on accurately estimating pseudo target labels might affect its efficacy when confronted with unusual data distributions or novel attack method.**

**Methods And Evaluation Criteria:**

The paper's methods and evaluation criteria are generally well-suited for addressing backdoor defense challenges, employing comprehensive experiments across popular attack methods (BadNets, Blend, SIG, and WaNet) and standard datasets (CIFAR-10/100, TinyImageNet). The evaluation framework effectively balances security and performance using ASR and accuracy metrics, while also introducing a useful DER or comparative analysis. The appendix D also includes a comprehensive hyperparameter analysis, and I believe that the evaluation are exceptionally robust.

**Other Comments Or Suggestions:**

The text font in Figure 1 is a bit small.

**Other Strengths And Weaknesses:**

The paper offers an innovative approach by creatively combining open-set recognition and loss-guided splitting techniques, thereby eliminating the need for extra clean data—a common and restrictive assumption in many earlier works. This originality is complemented by solid empirical evidence demonstrating its effectiveness across multiple benchmark datasets even under high poisoning rates and clean-label attack scenarios, highlighting its potential significance in real-world applications. The clarity of the experimental design and the thorough ablation studies are commendable, providing a well-documented analysis of the contributions of each module.  This is a good paper and I did not find any other weaknesses.

**Questions For Authors:**

- Could you elaborate on the rationale behind choosing the OSS and ALS perspectives over other alternatives such as gradient-based metrics, alternative feature distributions, or multi-modal representations? Have you considered or performed ablation studies comparing these different combinations?

- Could you provide more insight into the robustness of the target label(yt) estimation process? For instance, what are the failure modes if yt is misestimated, and can you quantify the impact on BSD’s segmentation and overall defense performance?

**Relation To Broader Scientific Literature:**

Prior research in backdoor defenses—such as ABL, DBD, and ASD—has largely depended on clean-data-dependent strategies or single-perspective methods (e.g., loss-guided splitting) to mitigate poisoning attacks. In contrast, the BSD integrates two complementary ideas from the broader literature. On one hand, it draws on open-set recognition techniques (similar to those used in OpenMAX) to harness semantic similarity and effectively distinguish benign samples from those that have been poisoned. On the other hand, it employs an altruistic model to capture differences in loss statistics, further refining the separation between clean and compromised data. By embedding these dual mechanisms within a semi-supervised learning framework inspired by MixMatch, the method overcomes the limitations of relying solely on extra clean data or a single detection perspective, thereby addressing issues observed under high poisoning rates and clean-label attack scenarios. This synthesis not only builds on established findings in robust deep learning and open-set recognition but also advances the state-of-the-art with a more scalable and cost-effective backdoor defense approach.

**Theoretical Claims:**

I reviewed the derivations of theoretical indicators in the paper—such as loss difference and open set distance—and found that these derivations rely primarily on intuitive explanations and ideas borrowed from existing literature rather than on a rigorous theorem-proof structure. Specifically, the paper employs the loss difference $I(x,y) = L_{\mathrm{sce}}(x,y,\phi) - L_{\mathrm{sce}}(x,y,\theta),$
and an Euclidean distance–based scoring function $S(x) = \min_{i \in \{0, 1, \dots, C-1\} \setminus \tilde{y}_t} \| f_e(x) - \mu_i \|_2$ to distinguish clean samples from contaminated ones. These methods are intuitively reasonable and consistent with previous work rather than strict mathematical proofs. Currently, the validity and robustness of the approach are mainly supported by extensive experimental results.

---

> ### Author Rebuttal · Authors · 2025-04-01
>
> Thank you for your thoughtful and constructive feedback. We appreciate your recognition of our contributions.
>
> We address your remaining concerns in the following sections.
>
> # Potential Failure of Target Label Estimation
>
> We would like to kindly emphasize that while target label estimation is an important part of our method's success in defending against backdoor attacks, we believe that based on existing literature and our experimental results, target label estimation is a relatively less challenging task compared to the defense itself. We have verified its robustness through extensive evaluations (Table 4, Figure 9, Table 8, Table 11). Moreover, target label estimation is not the primary contribution of our work. Our main contribution is the innovative reconstruction of the backdoor defense task into an Openset Recognition-like task.
>
> For potential failure cases, we have summarized four mitigation solutions:
>
> 1. In common scenarios where (1) the dataset is large and well-known with publicly available information on the number of samples per class, or (2) the dataset is well-balanced, target labels can be reasonably approximated using label statistics alone. This simple yet effective strategy ensures reliable target label estimation in practice.
> 2. We propose more than one target label estimation approach. Specifically, we introduce an alternative method based on local gradient ascent in Appendix D.3 (line 1012), which has proven effective against most of the discussed attacks. Furthermore, this alternative method may offer better target label estimation for unknown attack methods.
> 3. In scenarios where computational overhead is not a concern, we can apply the OSS algorithm to all possible classes and then aggregate the clean samples identified across all classes as the final OSS result. This approach has demonstrated effectiveness in our experiments (Figure 11).
> 4. For easy attacks, even if target label estimation fails, our pool update strategies can effectively correct the model through subsequent iterations (Figure 10).
>
> # The selection of OSS and ALS
>
> We sincerely appreciate your insightful comments. Our decision to adopt OSS and ALS was primarily driven by considerations of computational overhead, as well as the challenges posed by the non-clean subset and model-agnostic defense settings.
>
> Regarding gradient-based metrics, we agree that they can be effective. In fact, during the early stages of our research, we explored using per-sample gradient information and techniques like Grad-CAM to identify poisoned sample triggers. We further considered combining inpainting and denoising techniques in reverse diffusion to remove both local triggers (e.g., BadNets) and global triggers(e.g., Blended attacks). However, we ultimately abandoned these approaches as they significantly increased training time, particularly when performing adaptive pool updates.
>
> Regarding alternative feature distributions, most of them are designed for backdoor detection, often assuming a clean subset. However, this violated the clean-data-free scenario. Additionally, potential feature extraction from intermediate layers will undermine the model-agnostic feature of the algorithm. Defenders would require knowledge of the model structure to select intermediate layers, undermining the practicality of defenses.
>
> In contrast, our chosen OSS and ALS methods effectively integrate the defense task into the training process without making model-specific assumptions. Empirically, we observed that the OSS module, inspired by open-set recognition, complements the loss-based ALS, leading to robust defense performance.
>
> # Extreme hyperparameter situations
>
> We sincerely apologize if the statements(lines 434-439, page 8) caused concerns regarding the extreme hyperparameter scenarios. Our intention was to demonstrate the robustness of our method to the settings of the two key hyperparameters, $\alpha$ and$\beta$ . We believe the results presented in Table 7 provide a solid reference for choosing hyperparameters, offering practical guidance for defenders and future researchers. Specifically, both values can be set as floating-point numbers within the range of (0,1), with $0.3<\alpha<0.8$ and $\beta<0.5$ being relatively safe choices.
>
> To better clarify our points, we have revised the relevant section (lines 434-439, page 8) as follows:
>
> > We here present the influence of the main parameter, i.e., the parameters $\alpha$ and $\beta$ controlling the pool size. As revealed in Table 7, our BSD has robust performance against all the attacks with a relatively loose range of $\alpha$ and $\beta$ , and we recommend using the default setting in normal cases, and a reasonable range for adjustments is $0.3<\alpha<0.8$,$\beta<0.5$.
>
> # Minor issues
>
> Our adjustment allows the figure to be displayed at a larger scale on the page, see this anonymous link: https://postimg.cc/bsztqjQS.

---

### Decision · Program_Chairs · 2025-05-01

**Decision:**

Accept (poster)

**Comment:**

The paper received 2 Accept, 1 Weak Reject, and 1 Reject scores pre-rebuttal. After the rebuttal, only Reviewer FTAm changed his score from Reject to Weak Reject, after getting clearer on the paper's claims and settings.

Reviewers 2zkR and Mmmu applauded the paper for its interesting idea, clear writing, extensive experiments, and promising results. In contrast, Reviewers at apHg and FTAm found the writing unclear and required more clarification. Reviewer apHg questioned the method's effectiveness against clean-label attacks and the robustness of the target label estimation process. The rebuttal attempted to address his concerns but did not help to change the score. Reviewer FTAm found the novelty limited and required more comprehensive experiments.

The ACs checked the paper and the discussions. The rebuttal seemed to address the concerns raised by Reviewer apHg. As for the experiment comprehensiveness concern from Reviewer FTAm, we found the tested attacks (7) and defenses (6) sufficient in terms of both quantity and diversity. However, we agreed that the novelty was not strong and that the writing needed to be improved.

The ACs found the benefits of the paper outweigh its shortcomings. Hence, we agreed to accept the paper to ICML. The authors should consider the reviews to improve their camera-ready version.